# Transposable elements potentiate radiotherapy-induced cellular immune reactions via RIG-I-mediated virus-sensing pathways

Junyan Du[1], Shun-Ichiro Kageyama [2,3✉], Riu Yamashita[1,4], Kosuke Tanaka[5], Masayuki Okumura[2], Atsushi Motegi[2], Hidehiro Hojo[2], Masaki Nakamura [2], Hidenari Hirata [2], Hironori Sunakawa [6], Daisuke Kotani [7], Tomonori Yano[6], Takashi Kojima[7], Yamato Hamaya[1], Motohiro Kojima[8], Yuka Nakamura [8], Ayako Suzuki [4], Yutaka Suzuki [4], Katsuya Tsuchihara [1] & Tetsuo Akimoto[2,3]

Radiotherapy (RT) plus immunotherapy is a promising modality; however, the therapeutic effects are insufficient, and the molecular mechanism requires clarification to further develop combination therapies. Here, we found that the RNA virus sensor pathway dominantly regulates the cellular immune response in NSCLC and ESCC cell lines. Notably, transposable elements (TEs), especially long terminal repeats (LTRs), functioned as key ligands for the RNA virus sensor RIG-I, and the mTOR–LTR–RIG-I axis induced the cellular immune response and dendritic cell and macrophage infiltration after irradiation. Moreover, RIG-I-dependent immune activation was observed in ESCC patient tissue. scRNA sequencing and spatial transcriptome analysis revealed that radiotherapy induced the expression of LTRs, and the RNA virus sensor pathway in immune and cancer cells; this pathway was also found to mediate tumour conversion to an immunological hot state. Here, we report the upstream and ligand of the RNA virus sensor pathway functions in irradiated cancer tissues.

[1] Division of Translational Informatics, Exploratory Oncology Research and Clinical Trial Center, National Cancer Center, Chiba, Japan. [2] Division of Radiation Oncology and Particle Therapy, National Cancer Center Hospital East, Chiba, Japan. [3] Department of Radiation Oncology, National Cancer Center Hospital East, Chiba, Japan. [4] Department of Computational Biology and Medical Sciences, Graduate School of Frontier Sciences, The University of Tokyo, Chiba, Japan. [5] Division of Cancer Immunology, Exploratory Oncology Research and Clinical Trial Center, National Cancer Center, Chiba, Japan. [6] Department of Gastroenterology and Endoscopy, National Cancer Center Hospital East, Chiba, Japan. [7] Department of Gastroenterology and Gastrointestinal Oncology, National Cancer Center Hospital East, Chiba, Japan. [8] Division of Pathology, Exploratory Oncology Research & Clinical Trial Center, National Cancer Center, Chiba, Japan. ✉email: skageyam@east.ncc.go.jp

Radiotherapy (RT) is a widely used treatment for various cancers, including lung and oesophageal cancers[1]. Numerous recent reports have shown that irradiation can activate the antitumour immune reaction. It has been reported that RT induces the upregulation of type I IFN, MHC class I and PD-L1 (CD274) in cancer cells, and several immune cells, such as CD8 + T cells, dendritic cells (DCs) and M1 macrophages, are attracted to tissues after RT[2]. Thus, RT can induce the immunological conversion of cancer tissue to a hot state, and the induction of an antitumour immune reaction by RT is expected to be part of cancer therapy. One characteristic immunological effect of RT is the abscopal effect, in which local RT induces a systemic immune response in cancer patients[3,4]. Unfortunately, this effect is fairly rare and difficult to predict[5].

The most promising treatment involving immunotherapy and irradiation is the combination of immune checkpoint inhibitors (ICIs) and RT. As of 2021, combination treatment with ICIs and RT is a standard therapy for only non-small-cell lung cancer (NSCLC), although clinical trials are ongoing for almost all other types of solid cancers. Although interesting responses to RT and ICIs in phase I and II trials have been reported, most of these trials have reported negative or inconclusive results regarding patients who might benefit from such combination therapy[6]. Thus, RT with immunotherapy is expected to become an important strategy, but the outcomes are not yet adequate. Therefore, it is necessary to clarify the molecular mechanism of the RT-induced antitumour immune reaction to develop combination therapies comprising RT and immunotherapy.

The antitumour immune reaction after RT is a complicated event in the tumour microenvironment (TME) involving immune cells and stroma, and the underlying molecular mechanism is largely unknown. Events in cancer cells that occur in response to radiation are defined as the cellular immune response and are thought to trigger a radiation-induced antitumour immune reaction through IFN and cytokine release. Although several molecular mechanisms underlying this response have been reported, no consensus has been reached. In 2017, irradiation was reported to induce a DNA sensor-dependent cellular immune response, but the underlying mechanism was not clear: Harding et al.[7] reported a model involving the micronucleus–cyclic GMP-AMP synthase (cGAS)–stimulator of interferon genes (STING) pathway, whereas ref. [8] reported a STING-independent model in which a double-strand DNA break directly activates STAT1–IRF1. In 2020, Feng et al.[9] showed that the radiation-induced cellular immune response occurred in an RNA sensor-dependent manner, specifically via the polymerase III-dependent ncRNA–RIG-I–MAVS pathway, in the MCF10A. In contrast, ref. [10] reported in 2021 that mitochondrial RNA (mtRNA) is the key ligand in the RNA sensor pathway that dominantly regulates the RT-induced cellular immune response in MCF10 cells. Although such a different mechanism has been reported, there are some common features in the downstream pathways. For example, most papers identify the induction of a type I interferon (IFN) response via Y701-STAT1 phosphorylation in both the DNA and RNA sensor pathways as important[11,12]. In the RNA sensor pathway, melanoma differentiation-associated gene 5 (MDA5) and the retinoic acid-inducible gene I (RIG-I)/mitochondrial antiviral-signalling protein (MAVS) pathway are known to be responsible for sensing pathogenic RNA[11], and MAVS can recruit TANK binding kinase-1 (TBK1) to activate IRF3 and IRF7 to initiate the transcription of IFNs and proinflammatory cytokines[11,12]. In addition, the RIG-I–MAVS axis is known to be important in the radiation-induced immune response involving the RNA sensor pathway. However, various studies have not reached a consensus as to the ligand of RIG-I.

Although the molecular mechanism of the radiation-induced cellular immune response is being elucidated, little is known about the difference in dependence on the DNA or RNA sensor pathway or the ligand and upstream components of the RNA sensor. These differences are thought to depend on RT schedules and/or cancer origins[9]. Since NSCLC and oesophageal squamous cell carcinoma (ESCC) are major cancers for which definitive RT is indicated, and clinical trials of RT + ICI combination therapy in these indications are expected, we focused our analysis on NSCLC and ESCC.

In this study, we found that the RNA virus sensor pathway involving RIG-I was significantly upregulated after irradiation in the A549 NSCLC cell line. We also identified long terminal repeats (LTRs), a group of transposable elements (TEs), as ligands for RIG-I. LTRs have been reported to be induced by chemotherapy and involved in ICI resistance[13–16], but have not been reported with RT. We aimed to elucidate whether this LTR–RIG-I axis regulates both the cellular immune response and the antitumour immune reaction by in vitro cell line experiments and ex vivo experiments using human peripheral blood mononuclear cells (PBMCs) and the analysis of cancer tissue from patients undergoing RT.

## Results

**Radiation induces an innate immune response via a viral response pathway**. To investigate the cellular immune response to irradiation in A549 LUAD epithelial cells, we first investigated the dynamics of immune responses after 8 Gy irradiation and then screened regulators at particular time points by proteome analysis. We performed ISRE reporter assays (Fig. 1a) after irradiation, as ISRE activity has been reported to be a surrogate for the cellular immune response[17]. IR A549 cells were sampled at 48 h, 96 h, 7 days and 11 days after 8 Gy irradiation, and NIR cells were analysed at the same time points as a control. ISRE activity was significantly increased 48 h after irradiation and continued to increase in a time-dependent manner until day 7 (Fig. 1b). We further confirmed the cellular immune response at the protein level; p701-STAT1 and PD-L1 expression levels were consistent with ISRE activity (Fig. 1c).

To identify the signalling pathway that regulates the radiation-induced cellular immune response, we performed a phosphoproteomic analysis of IR and NIR cells at 48 h post-irradiation, the time point at which ISRE activity began to significantly increase. The peptides with more than a twofold change in abundance in IR cells compared with NIR cells (Fig. 1d) were enriched in GO biological process (BP) terms; the top 15 BP terms were identified (Fig. 1e). The top category was related to viral genome replication, and the pathways in this category have well-known essential roles in sensing RNA viruses and triggering the IFN response[18], such as MAVS, OAS3, and MDA5 (IFIH1) (Fig. 1 f). It has been reported that whether the immune response to radiation is DNA sensor dependent or RNA sensor dependent differs from cell to tissue. Although it is unclear whether A549 cells depend on DNA sensors or RNA sensors, we hypothesised that the cellular immune response in A549 cells and other LUAD cells would be dependent on RNA sensors, especially RNA virus sensors.

Next, we confirmed that the RNA virus sensor could regulate ISRE activity, the type I IFN response and immune cell infiltration in LUAD tissues. We performed a correlation analysis using TCGA datasets to investigate the relationship of RNA virus sensor pathways with tumour immune responses in NSCLC patient tissues. Compared with that of the DNA sensors cGAS and IFNγ, which are known regulators, the expression of the RNA sensor genes RIG and MDA5 was more strongly correlated

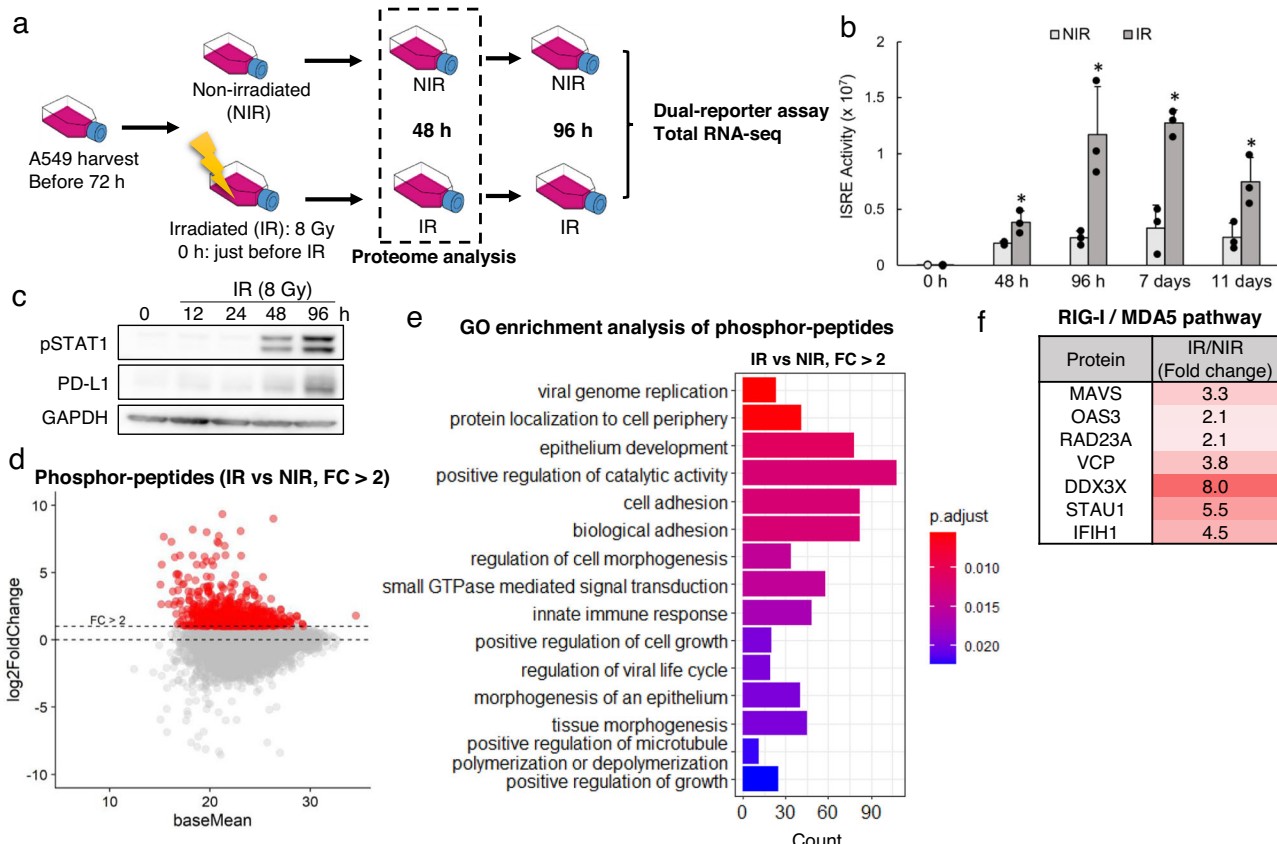

**Fig. 1 Radiation induces an innate immune response via a viral response pathway. a** Experimental design. **b** Interferon-stimulated response element (ISRE) activity in A549 cells was determined using luciferase assays at 0, 48, 96 h, 7 d and 11 d after no irradiation (NIR group) or 8 Gy irradiation (IR group). Data were presented as the mean ± SEM of three biological replicates; *$p \leq 0.05$ and **$p \leq 0.01$ compared to the NIR group at 0 h, paired $t$-test. ISRE activity in A549 cells was determined using a luciferase assay following IR. Error bars represent the SEM of three biological replicates. *$p \leq 0.05$ and **$p \leq 0.01$, paired $t$-test. NIR nonirradiated, IR irradiated. **c** Representative western blots ($n = 3$ independent experiments) of activated (phosphorylated) STAT1 and PD-L1 (CD274) at the indicated times after 8 Gy irradiation. GAPDH served as the loading control. **d** Phosphorylation dynamics of peptides with or without irradiation, as determined using Scaffold DIA. The red dots indicate peptides with more than a twofold change in abundance in IR cells compared with NIR cells. **e** Gene Ontology (GO) enrichment analysis of the phosphopeptides (the red dots) shown in **d**. X-axis represents the count of differentially expressed genes (DEGs) belonging to each GO term. **f** Differential phosphorylation of viral response-related proteins before and after irradiation.

with the expression of OAS2, MX1 and OASL, which have ISRE promoters (Supplementary Fig. 1a). This correlation of RIG-I/MDA5 with ISGs in LUAD tissue confirmed our aforementioned hypothesis that the immune response was associated with RIG-I/MDA5—MAVS in this context. Although the DNA sensor pathway is currently considered the standard pathway for cancer immune responses, our results indicate that ISRE activation may be RNA virus sensor-dependent in A549 cells and LUAD tissues.

IFNB, IRF7 and IRF9, which are downstream of the RIG-I pathway, and the infiltration of macrophages, DCs, neutrophils and CD8 + T cells also correlated with RIG-I expression in LUAD (Supplementary Fig. 1a, b). These results indicate that the RNA sensor pathway may dominantly regulate ISRE activity and immune cell infiltration in LUAD tissue.

Next, we confirmed that irradiation induces RNA virus-like ncRNA and RNA virus sensor pathway activation in A549 cells. A549 cells exposed to 8 Gy irradiation showed a significant increase in virus-like dsRNA compared to that in NIR cells (Fig. 2a, b). By total RNA-seq, we detected a subset of LTRs that were upregulated at 7 days after irradiation (Fig. 2c). Conversely, the changes in short interspersed elements (SINEs), long interspersed elements (LINEs) and small ncRNAs were subtle (Supplementary Fig. 2a–c). Interestingly, we also identified several specific LTRs, including LTR21B and MER57F (Fig. 2d),

which have been reported to strongly correlate with the local tumour immune response[19], to be elevated after irradiation. We also confirmed the protein expression levels of RIG-I, IRF7 and pY701-STAT1 (Fig. 2e), which have been reported to be downstream of RNA virus response genes in A549 cells[20]. We conclude that irradiation induces the expression of virus-like ncRNAs, including LTRs, at the RNA and protein levels in A549 cells; these LTRs then activate the RNA virus sensor pathway.

**Identification of RIG-I ligands by RIP-seq.** Our results revealed the RNA sensor dependency of ISRE activity in both LUAD tissue and A549 cells, so we aimed to identify the RNA virus sensor ligand; we aimed to verify that radiation-induced LTRs act as ligands in the RNA sensor pathway by identifying RIG-I-bound RNA after the pulldown of FLAG-tagged RIG-I (Fig. 3a). RNA bound to RIG-I in IR cells was recovered and identified by total RNA-seq as the RIG-RIP group, whereas total RNA in the cell lysate was sequenced as the input group. TEs such as DNA, LINEs, LTRs, and SINEs were significantly enriched in the RIG-RIP sample, but LTRs were the most prominently enriched species. As LTR21B has been reported to correlate with local tumour immune responses[19,21], although it is not clear that LTR21B is the main ligand in radiation-induced immune reaction, we chose to focus subsequent experiments on this specific LTR as an

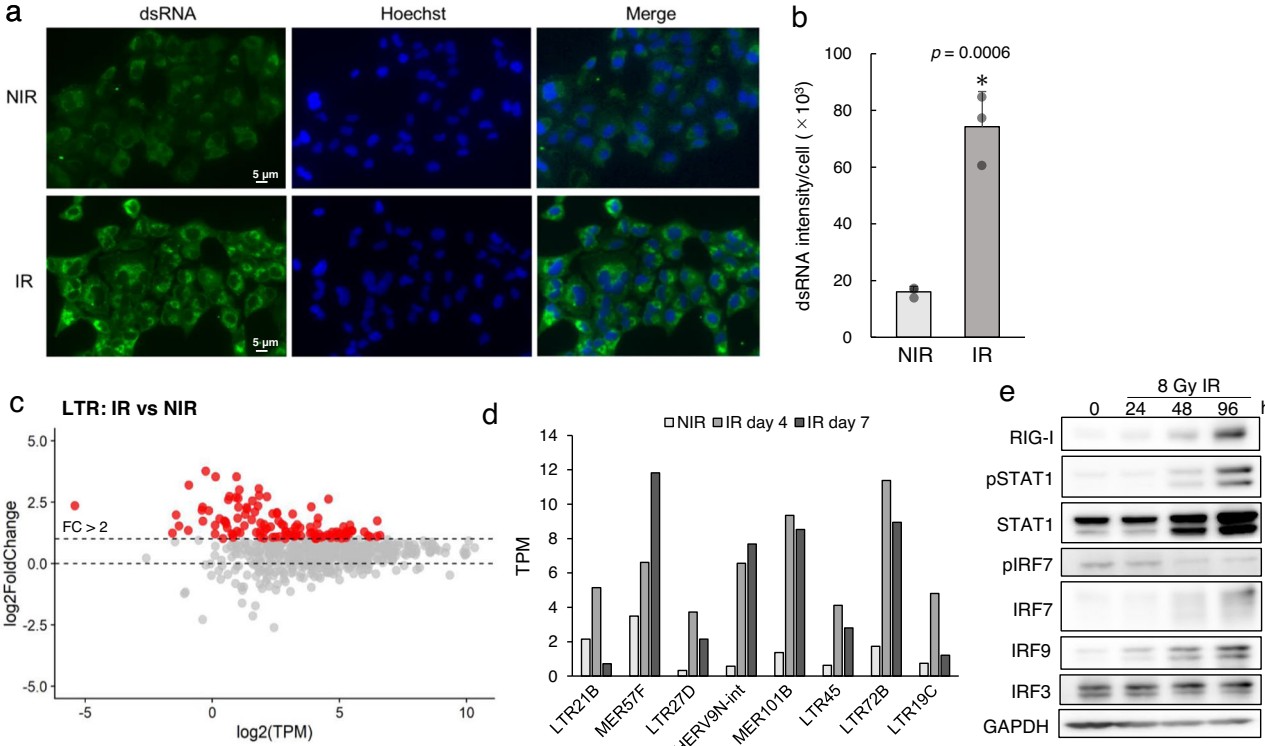

**Fig. 2 dsRNA and ERV levels increase after irradiation. a** Immunofluorescence staining for K1 antibody indicating dsRNA, indicative of residual DSBs in A549 cells. Scale bars: 20 and 2 μm for the magnified images. **b** Quantification of dsRNA at 3 h after irradiation. Positive areas and signal intensities were automatically calculated using a hybrid cell count application (BZ-H4C, KEYENCE). Data were presented as the mean ± SEM of three biological replicates. *$p \leq 0.01$, paired $t$-test. **c** Fold change in LTR expression between irradiated (IR) and nonirradiated (NIR) A549 cells. The IR group was harvested 7 days after 8 Gy irradiation, and the NIR group was harvested at the same time without irradiation. The expression of LTRs was quantified by total RNA-seq. Data were inclusive of two independent experiments. The red dots indicate the LTRs with a more than a twofold change in abundance in IR cells compared with NIR cells. **d** Immune-related LTR expression levels in the NIR group and in the IR group at 4 and 7 days post-irradiation. **e** Protein levels of RIG-I and representative markers of immune activation.

indicator. We confirmed that LTR21B causes ISRE activation similar to irradiation in a time-dependent manner; this response was dependent on RIG-I. Transfected LTR21B significantly induced ISRE activity from 24 to 48 h (Fig. 3d), similar to the effect of irradiation. ISRE activity was not induced by LTR21B in RIG-I-KO cells (Fig. 3e), suggesting that the activity of LTR is RIG-I dependent in A549 cells.

**High-throughput screen (HTS) of regulators of the irradiation-induced RNA sensor pathway**. We aimed to identify the pathway upstream of the radiation-induced LTR elevation discovered in this study by using a previously reported HTS[17] and kinase inhibitor library[22]. A549-Dual™ cells were treated with kinase inhibitors ($n = 798$; final concentrations: 5, 0.5 and 0.05 μM) 1 h before 8 Gy irradiation, and luciferase activity, indicative of ISRE activity, was measured 96 h after irradiation (Fig. 4a). At a concentration of 0.5 μM, 137 kinase inhibitors showed at least 50% suppression of IRSE activity, with 32 suppressing IRSE activity by 80% or more (Fig. 4a).

The irradiation-induced immune response was reported to be increased by the ATR pathway inhibitor AZD6738 but decreased by the CDK and JAK inhibitors dinaciclib and decemotinib, respectively[7,9]. These results were confirmed in our study (Fig. 4b), indicating the effectiveness of our HTS. In addition, we found that the mTOR inhibitor WYE-125132 strongly suppressed ISRE activity (Fig. 4b). In contrast to the effects of direct mTOR inhibition, inhibition of either PIK3, an mTOR activator, or autophagy, the main biological target of mTOR, had

limited effects on ISRE activity. In particular, multiple mTOR inhibitors suppressed ISRE activity, even at low concentrations of 50 nM (Fig. 4b and Supplementary Fig. 3a). We focused on the specific inhibition of ISRE activity by this mTOR inhibitor, although the mechanism independent of PIK3 and autophagy is unknown.

The mTOR kinase has multiple targets, and we wanted to confirm whether mTOR suppresses broad pathways, such as protein synthesis, or specific factors, such as ISRE activity or the RNA virus sensor pathway, in A549 cells. To identify the mTOR target pathway in A549 cells after irradiation, GO term analysis and phosphoproteome analysis were performed for the IR vs. IR plus mTOR inhibitor WYE-125132 groups. GO term analysis identified the viral replication pathway as the GO term most significantly enriched in the targeted group of genes (Fig. 4c). Thus, we next aimed to identify potential direct targets of mTOR by phosphoproteome analysis. In this analysis, we first confirmed the phosphorylation of two genes, the DNA damage marker CHEK2, the phosphorylation of which is independent of mTOR, and the immune response marker STAT1, as positive controls; we also analysed MTOR. Phosphorylated CHEK2, STAT1 and MTOR levels increased in response to irradiation, but STAT1 and MTOR phosphorylation decreased after mTOR inhibition and irradiation (Supplementary Fig. 3b). These protein dynamics indicated reasonable performance of the analyses.

Next, we investigated genes that have been reported to be involved in LTR expression, stability and sensing, and among these genes, we identified several genes whose phosphorylation was increased by radiation and decreased by mTOR inhibition as

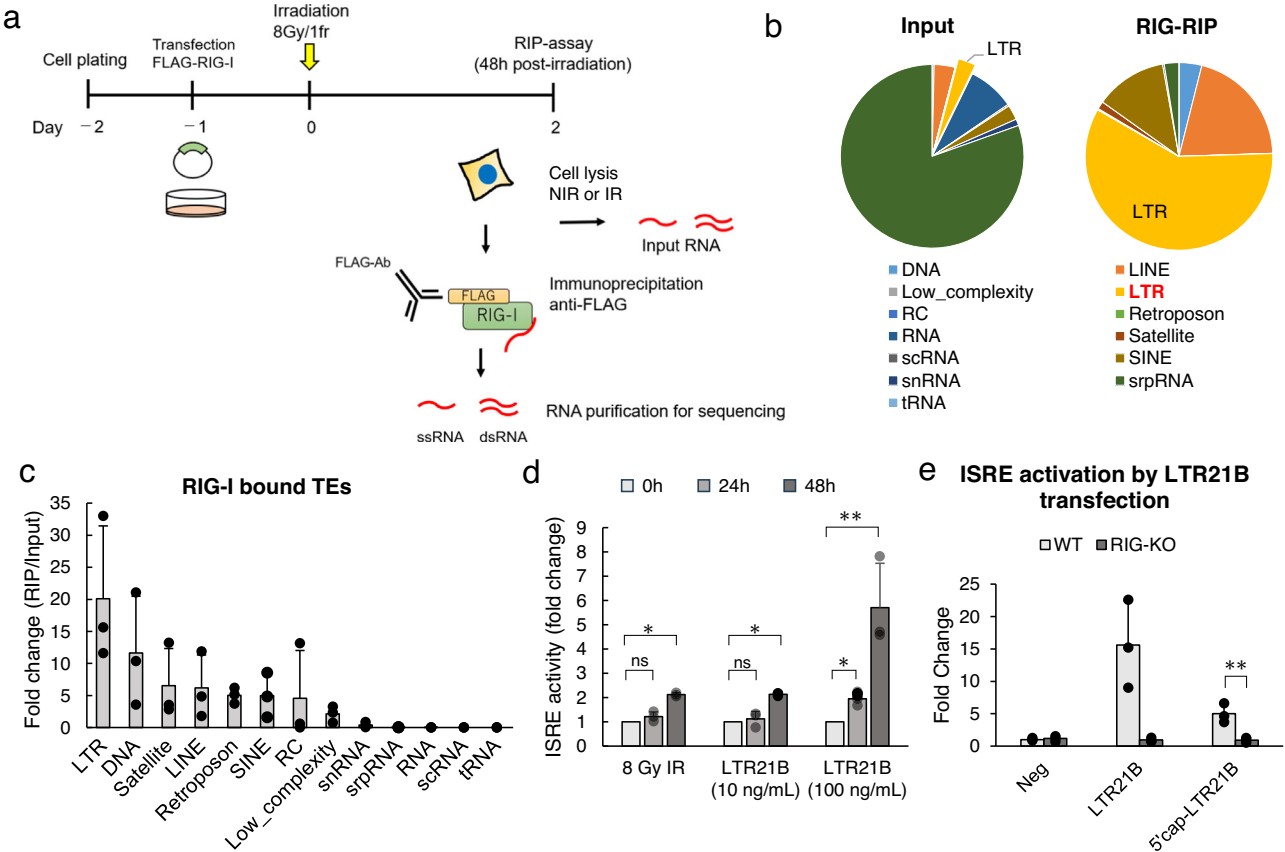

**Fig. 3 Identification of RIG-I ligands by RNA immunoprecipitation (RIP)-chip assay. a** Experimental design. **b** Proportion of transposable elements (TEs) enriched in the input and RIG-RIP samples. Proportion = TPM sum of the indicated TE family/TPM sum of all TEs. Data were from three biological replicates. **c** Fold change in expression for each TE family. Fold change = TPM sum of the indicated TE family in the RIG-RIP sample/TPM sum of the indicated TE family in the input sample. Error bars represent the SEM of three biological replicates. **d** Quantification of ISRE activity in A549 cells at 0, 24 and 48 h after transfection with the indicated concentration of synthesised LTR21B. Error bars represent the SEM of three biological replicates. ns no significance; *$p \leq 0.05$ and **$p \leq 0.01$ compared to 0 h by paired $t$-test. **e** Quantification of ISRE activation in WT and RIG-KO A549 cells 48 h after LTR21B transfection (100 ng/mL). Error bars represent the SEM of three biological replicates. *$p \leq 0.05$ and **$p \leq 0.01$, paired $t$-test.

potential mTOR targets. NUCKS1 is involved in LTR expression, DDX3X and STAU1 are involved in LTR stabilisation, and IFIH1 (MDA5) are RNA sensors[23–26]. Unfortunately, the biological significance of most of the phosphorylation sites identified in this study has not been determined, and there are multiple candidates, so direct targets could not be identified in this study.

Finally, we performed a protein knockdown experiment to confirm the specificity of mTOR inhibition. We aimed to establish cells with a genetic knockout of mTOR1 exon 1 and identified two clones with a genetic knockout of both alleles; however, weak mTOR protein expression from the second (downstream) ATG codon was observed (Supplementary Fig. 3d). Therefore, we defined these mTOR-KO cell clones to have genetic knockout and stable protein knockdown. The mTOR-KO cells exhibited significantly reduced radiation-induced ISRE activity similar to that of mTOR inhibitor-treated cells, and this reduction was not rescued by autophagy inhibition (autophinib or ULK101) (Supplementary Fig. 3e, f). In contrast, RNA-seq revealed a reduction in radiation-induced TE activation in mTOR-KO cells compared to WT cells (Fig. 4d). We also confirmed autophagy independence in the mTOR-KO clones, as observed in the inhibitor experiment (Supplementary Fig. 3c, e, f).

Our results indicate that the mTOR pathway specifically activates the RNA virus sensor pathway after irradiation and that the mTOR-dependent increase in LTR levels is a novel mechanism, although it remains unknown whether the increase is due to expression or stabilisation.

**RIG-I/MDA5 knockout diminishes irradiation-induced cellular immune responses**. We confirmed the functions of RIG-I, MDA5 and MAVS in the cellular immune response to irradiation in A549 cells. Although ISRE activity is an important surrogate, the genes that directly function in the antitumour immune reaction are also important. For example, the MHC class and CD274 work in cancer and immune cells, and CCL5, CXCL10 and IFNB1 are downstream of ISRE and induce immune cell infiltration into cancer tissue.

We investigated whether RIG-I and MDA5, components of the RNA sensor pathway, regulate IFNB, CCL5, CXCL10, HLA-B and CD274 expression in cancer tissue using the same TCGA LUAD dataset described above. The expression of the RNA sensor genes RIG-I and MDA5 was significantly correlated with that of OAS2, HLA-B, IFNB1, CD274, CCL5 and CXCL10 (Fig. 5a). Moreover, OAS2, HLA-B, and CD274 levels were upregulated by irradiation in parallel with RIG-I and MDA5 levels (Fig. 5b) in the cellular immune response after irradiation in A549 cells.

This result indicates that the RNA virus sensor pathway is a dominant regulator of not only the ISRE but also these immune responses. Knockout of RIG-I, MDA5, or MAVS significantly reduced radiation-induced OAS2, IRF9, IRF7, HLA-B and CCL5 expression (Fig. 5c) and ISRE activity (Fig. 5d). The expression of CXCL10 and IFNB1 was similarly sensitive to RIG-I, MDA5, or MAVS knockout (Supplementary Fig. 4a), but the reductions

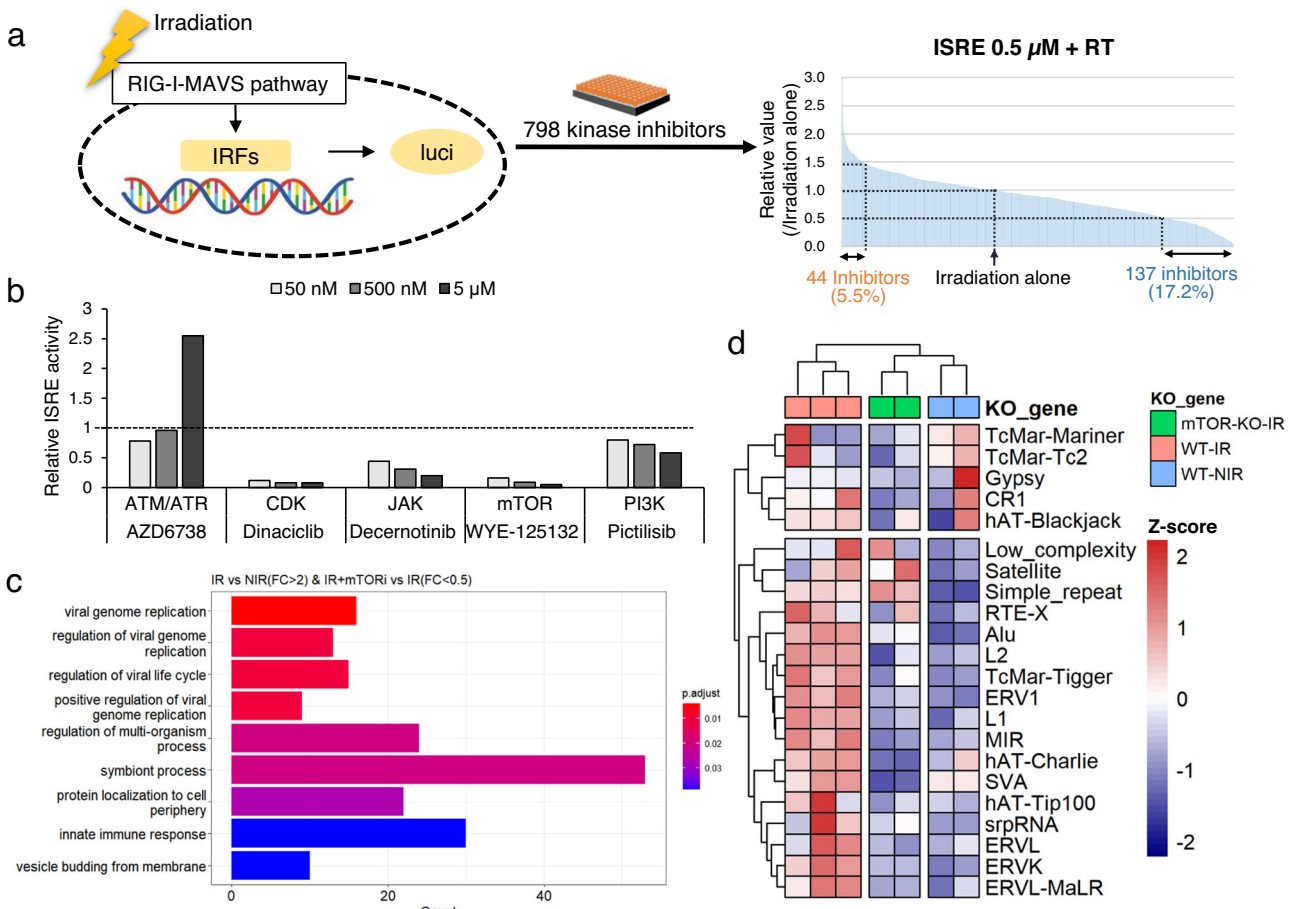

**Fig. 4 High-throughput screen (HTS) of regulators of the IR-induced RNA sensor pathway. a** Experimental design (left) and the results (right) of the HTS of the kinase library. At a concentration of 0.5 µM, 137 kinase inhibitors suppressed IRSE activity by at least 50%, with 32 suppressing IRSE activity by 80% or more. **b** ISRE activity in A549 cells at 96 h after treatment with the indicated inhibitors at the indicated concentrations. Data were representative of two independent experiments. **c** GO terms enriched in genes whose expression was induced by radiation (fold change >2) and suppressed by mTOR inhibition (fold change <0.5). **d** Expression of TEs in WT and mTOR-knockout (KO) IR cells and in WT NIR cells. Data were representative of biological replicates. The Z-score was calculated using the R package pheatmap from the TPM of the indicated TEs, and then clustering was performed using the same package.

were not statistically significant. This result indicates that the RNA sensor gene regulates not only the expression of the MHC class and CD274, which function in cancer cells, but also the cytokines that attract surrounding immune cells.

We also investigated the effects of RIG-I, MDA5 and MAVS knockout on broad immune gene expression patterns and TEs by RNA-seq. RIG-I knockout and MAVS knockout evoked the same transcriptional pattern, and the data clustered together, but MDA5 knockout did not (Fig. 5e). This result suggests that the RIG-I–MAVS pathway is directly involved in RT-induced immune responses and that MDA5 partially functions in these responses. The effect of gene knockout was also confirmed at the protein level (Supplementary Fig. 4b). Furthermore, knockout of RIG-I, MDA5, or MAVS had a mild effect on TE activation (Supplementary Fig. 4c). This result supports our hypothesised cascade: mTOR induces the upregulation of LTRs, and LTRs activate the RIG-MAVS pathway.

Most cancer immune responses are carried out by infiltrated immune cells among cancer cells. Type I IFN has been reported to attract immune cells, including lymphocytes and DCs, and our results also show that the RNA virus sensor pathway regulates IFN and cytokine expression to attract immune cells to LUAD tissue and A549 cells. Therefore, the effects of cellular immune reactions on PBMCs were investigated in terms of the absolute

number of infiltrated cells, activation status based on gene expression and cell population among PBMCs in ex vivo experiments. Infiltrated PBMCs were significantly increased in irradiated A549 cells, and IR RIG-I knockout significantly reduced the number of PBMCs (Fig. 6a, b). Furthermore, we performed mRNA-seq to confirm the activation status and cell population among the migrated PBMCs. The migrated PBMCs were collected 17 h after irradiation and identified by xCell[27]. The levels of cytokines such as IL1a, IFNB1, CD80, CD86, and CXCL10, which are important for inducing an immune response by macrophages within cancer tissue[28,29], were upregulated in PBMCs in a RIG-I-dependent manner (Fig. 6d and Supplementary Fig. 5a). The major cell types whose abundance was increased in A549 cells with IR and decreased with RIG-I KO were DCs and macrophages (Supplementary Fig. 5b, c). This infiltrating cell profile in the early stage after IR corresponds with previously reported data; notably, a recent scRNA-seq study reported that DCs and macrophages can work with CD4 + CD8 + T cells[30,31]. We suggest that DC and macrophage activation leads to antitumour reactions after RT.

**The radiation-induced LTR–RNA sensor pathway in oesophageal cancer tissues and cell lines.** Finally, we investigated in

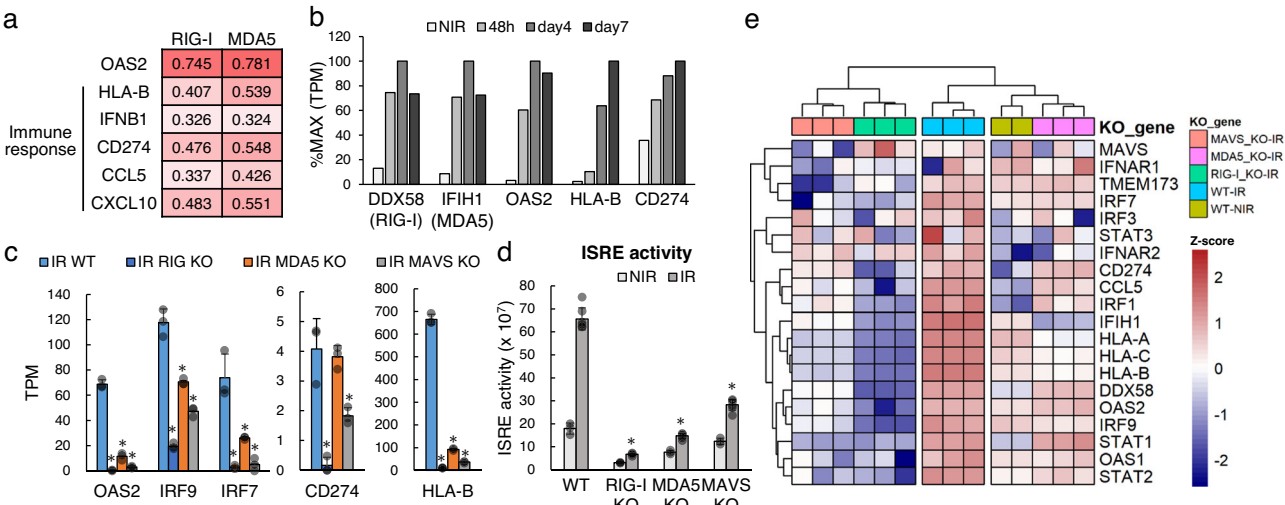

**Fig. 5 RIG-I and MDA5 knockout diminish the IR-induced cellular immune response. a** Spearman's correlation analysis of the expression of RIG-I and MDA5 with that of representative genes related to the immune response (OAS2, HLA-B, IFNB1, CD274, CCL5 and CXCL10) in TCGA lung adenocarcinoma sample sets (Firehose Legacy, $n = 584$). **b** Expression dynamics of DDX58 (RIG-I) and immune response markers in nonirradiated (NIR) A549 cells and irradiated (IR) cells exposed to 8 Gy radiation. **c** Expression of representative immune-active markers in wild-type (WT), RIG-I knockout (KO), MDA5-KO and MAVS-KO cells. Error bars represent the SEM of three biological replicates. *$p \leq 0.05$ compared to the IR_WT group, Student's $t$-test. **d** Quantification of ISRE activity in WT and KO A549 cells after radiotherapy (RT). Error bars represent the SEM of three biological replicates. *$p \leq 0.05$, Student's $t$-test. **e** Expression of representative immune-active genes in NIR WT cells and in IR WT and KO cell lines. The $Z$-score was calculated using the R package pheatmap from the TPM of the indicated TEs, and then clustering was performed using the same package.

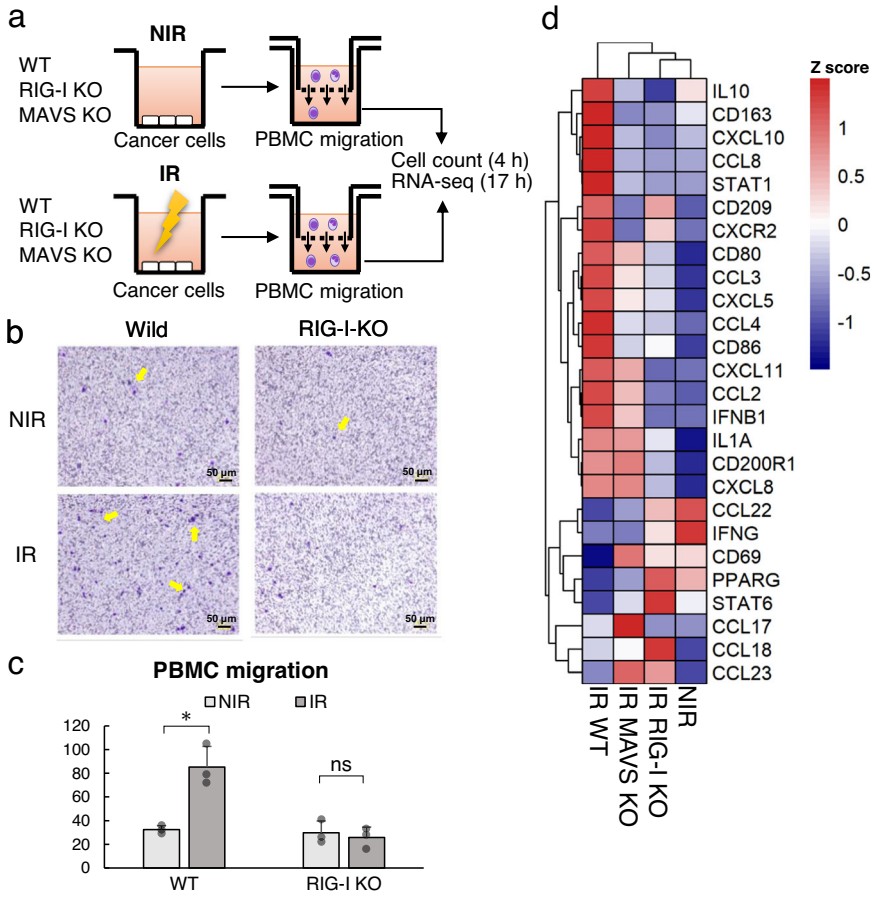

**Fig. 6 RIG-I knockout significantly reduces PBMC migration. a** Experimental design. **b, c** Staining of PBMCs (**b**) and quantification of PBMC migration (**c**). Scale bar: 50 μm. Error bars represent the SEM of three independent experiments. *$p \leq 0.05$ compared to NIR group cells, Student's $t$-test. WT wild-type, KO knockout, IR irradiated NIR nonirradiated. **d** Expression of representative markers of immune activation assessed by RNA-seq in PBMCs cultured with medium from the indicated cells. Data from a single experiment are shown.

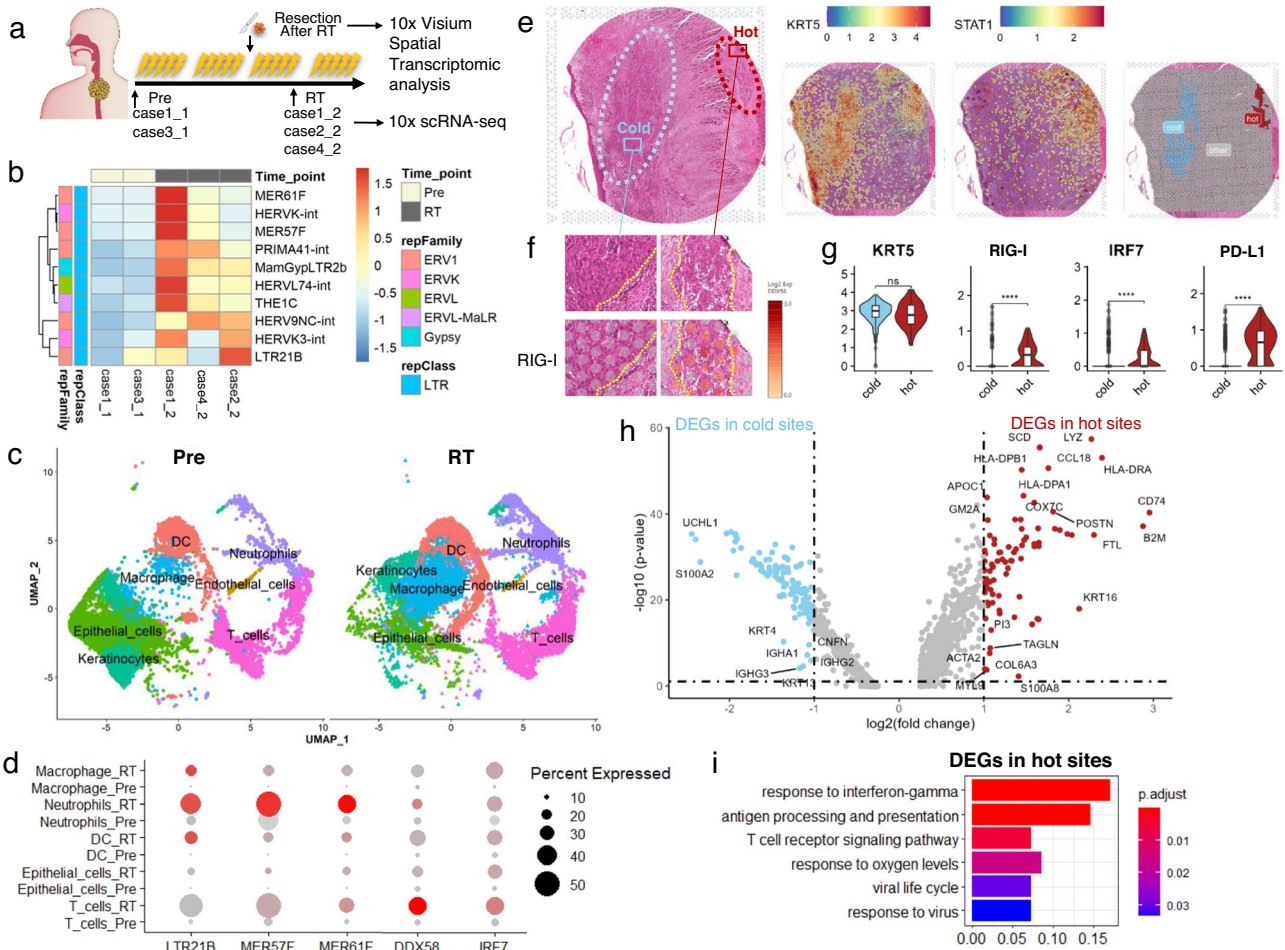

**Fig. 7 Radiation-induced LTR–RNA sensor pathway in oesophageal cancer tissues. a** Experimental design. Pre, before radiotherapy; RT, after radiotherapy. **b** Expression of immune-related LTRs in oesophageal tissue, as determined by scRNA-seq. The heatmap is based on the average positive ratio of the indicated LTRs in each sample. **c** UMAP of cell clusters in the Pre and RT samples. **d** Bubble chart of the percentage of cells within the indicated cell cluster expressing the indicated LTRs and immune-active marker genes and the average gene expression level. A larger dot indicates a higher percentage of cells expressing a particular LTR/gene, and a darker dot colour indicates a higher average expression level of the LTR/gene. **e** Morphology of the 10x Visium slide (left), spatial expression of the epithelial marker KRT5 (middle-left) and the immune-active marker STAT1 (middle-right), and manual annotation of hot and cold sites (right). **f** Morphology of hot and cold sites. **g** Expression of KRT5, RIG-I, IRF7 and PD-L1 in hot and cold sites (****$p < 1 \times 10^{-16}$, Bonferroni-adjusted Wilcoxon test; ns no significance). **h** Differentially expressed genes (DEGs) between hot and cold sites. DEGs were defined as those for which $p < 0.05$ (Bonferroni-adjusted Wilcoxon test) and fold change >2. **i** GO enrichment analysis of the DEGs identified in hot sites.

detail whether RT activates transposons and RNA virus sensors in patient tissues within the TME. As it is difficult to obtain tissue samples by endoscopy from patients with lung cancer undergoing RT, oesophageal cancer is one of the easiest cancers in which to apply endoscopy, and our previous studies have shown that an RNA sensor-dependent cellular immune response is induced in oesophageal cancer cell lines (Supplementary Fig. 6a).

An outline of our tissue analysis is shown in Fig. 7a. In all cases of conventional RT with 1.8–2 Gy irradiation carried out once a day, a biopsy is performed pre and after RT to obtain a cell-by-cell transcriptome, and we first confirmed activation of the LTR and RIG-I pathways in cancer cells. Furthermore, changes in normal cells, such as immune cells, were also investigated. We performed scRNA-seq in tumour tissues collected by endoscopy from oesophageal cancer patients before (Pre, $n = 2$) and during RT ($n = 3$) (Fig. 7a and Supplementary Table 1). First, we integrated the 5 samples for unsupervised clustering analysis and identified 13 distinct cell clusters (Supplementary Fig. 7a, b), each of which was annotated by using SingleR (Supplementary Fig. 7c) and representative cell markers (Supplementary Fig. 7d). The identified T-cell, DC, macrophage, neutrophil, endothelial cell,

and keratinocyte clusters were selected for comparison analysis before and during RT (Fig. 7b, c and Supplementary 7e, f). At the single-cell level, the levels of ERVs, which include LTRs, were upregulated during RT (Fig. 7b). The levels of specific LTRs, such as LTR21B, MER57F, DDX58 (RIG-I) and IRF7, were also upregulated in epithelial cells, which included ESCC cells and other normal immune cells (Fig. 7d). Taken together, the data indicate that LTR–RIG pathway upregulation generally occurs in the TME, in not only cancer cells but also normal immune cells, during RT.

To better understand the process by which the RIG-I pathway is upregulated in the tumour immune reaction to RT, surgically resected oesophageal cancer tissue from one patient who received 41.4 Gy/23 fr RT was subjected to spatial transcriptome analysis. A total of 5000 spots were captured, sequenced and grouped into 17 clusters (Supplementary Fig. 8a). The remaining cancer tissue was divided into two categories: hot sites containing damaged cancer cells and considerable immune cell infiltration and cold sites containing dense, viable cancer cells and little immune cell infiltration (Fig. 7e, f). The morphologically identified hot and cold sites were confirmed by marker gene analysis (Fig. 7g and

Supplementary Fig. 8b, c) and DEG analysis (Fig. 7h). There were significant differences between the hot and cold sites in the expression of immune cell activation surrogates, such as STAT1 and HMGB1, and immune cell infiltration indicators, such as the T-cell markers CD4 and CD8a, the macrophage marker CD68, and the DC marker HLA-DRA. However, the hot and cold sites did not differ in the levels of the epithelial marker KRT5 (Supplementary Fig. 8c, d and Fig. 7f, g). Interestingly, the hot and cold sites of the immune response were clearly separated in the post-radiation tissue, and spatial transcriptome analysis was able to distinguish between these two parts. The levels of DDX58 (RIG-I), IRF7, CD274, and the ERV gene family member ERV3-1 were upregulated in the hot sites of the ESCC tissue sample (Fig. 7g and Supplementary Fig. 8d). In total, 82 DEGs were identified in the hot sites, including immune-active genes such as CD74 and B2M. Then, to confirm virus sensor activation in cancer cells, we performed a GO enrichment analysis of the identified hot-site DEGs. The enriched terms included both those related to the immune response, such as the response to IFNγ the T-cell receptor signalling pathway, and antigen processing and presentation, and those related to viral activity and response, such as the viral life cycle and response to viruses (Fig. 7i). Thus, our spatial transcriptome analysis revealed that RT induces the formation of immunologically hot sites in regions of ESCC tumours containing cancer cells, in which the ERV3-1, DDX58, and virus response pathways are upregulated. Thus, scRNA-seq and spatial transcriptome analysis clearly revealed the upregulation of the LTR and RIG-I pathways by RT in the TME in human tissue.

## Discussion

To clarify the molecular mechanism of radiation-induced anti-tumour immune reactions, we conducted in vitro and ex vivo with PBMCs experiment and patients tissue analysis.

The most important discovery of this study was that TEs and the protein RIG-I were induced by 8 Gy irradiation, and RIG-I showed a strong interaction with the RT-induced LTRs. LTR upregulation is sufficient to activate immune responses, including the type I IFN response and DC and T-cell infiltration, in cancer tissue[14]. However, it is not clear whether an overall increase in LTRs or an increase in a specific LTR or type of LTR is important for the antitumour immune reaction. Several specific sequences containing LTR21B and MER57F were reported to be important for activating the immune response in most solid cancers[19]. We found that irradiation induced the expression of several LTRs, including LTR21B and MER57F, and that LTR21B was sufficient to induce ISRE activity in a RIG-I-dependent manner in A549 cells. Although there have been numerous reports in recent years that LTRs regulate the antitumour immune reaction, the induction of LTRs by RT has not yet been reported. This study is the first to report that radiation increases LTR levels and activates ISREs through the RIG-I–MAVS pathway.

Importantly, the identified mechanism was triggered by a clinically relevant radiation dose. The upregulation of TEs by irradiation has been reported previously, but those previous studies utilised irradiation with less than 2 Gy or chronic exposure to radiation as an environmental stressor and mainly investigated the potential activation of DNA elements such as SINEs and LINEs[32,33]. As it is common to administer doses of 8 Gy or higher to patients undergoing cancer treatment, it is important to understand the biological pathways activated by these higher doses in the context of clinical oncology; the predominant activation of LTRs among TEs was characteristic of high-dose radiation. Although snRNAs, LTRs, AT-rich RNA, and mtRNAs have been reported as ligands for RNA sensors in the

radiation-induced immune response, only TEs were identified as ligands in the RIP assay in our study[9,11,34]. This discrepancy among studies may stem from differences in the radiation dose and schedule or cancer type. We confirmed that the LTR–RIG pathway was activated in tissue from a patient given conventional RT (1.8–2 Gy once daily for a total of 20–30 fractions). As this schedule is the most frequently used for RT in NSCLC and ESCC, which are major cancers for definitive RT, we concluded that the LTR–RIG pathway might be a common regulator of the cellular immune response during definitive RT.

Our study inspires many hypotheses and questions. LTR induction has been reported to be induced by a variety of cancer treatments[13]. Chemotherapy with compounds that affect chromatin, such as DNMT and HDAC inhibitors, activates TEs through the reversal of histone silencing[13,35], as TEs are generally inactivated by chromatin modification via H3K9me3 and DNA methylation[36]. However, radiation does not directly affect H3K9me3, and little is known about the factors upstream of the RNA sensor pathway that are affected by irradiation. Therefore, we hypothesise that radiation-induced LTR activation is regulated by a pathway independent of epigenetic modification.

mTOR is a multifunctional kinase that directly regulates many biological processes, including hypoxia, nutrient starvation and DNA damage[37,38], and the mTOR pathway was identified as a candidate inhibitor of the irradiation-induced immune response. We initially hypothesised that mTOR inhibition would suppress the immune response through the activation of autophagy, as previously described in ref. [12]; however, our results led us to reject this hypothesis. Recent reports stated that irradiation induces mTOR activation[39] and that mTOR inhibition suppresses virus replication in vitro and in vivo[40,41]. We found that inhibition of the mTOR pathway resulted in the downregulation of the viral replication pathway, LTR expression and the phosphorylation of proteins associated with viral replication. However, we did not identify a direct target of mTOR in this study, and future research is needed to build on the novel discovery of upstream factors in the RNA sensor pathway by elucidating the mechanism by which mTOR elicits LTR upregulation.

scRNA-seq analysis revealed that during RT, the levels of LTRs, DDX58 (RIG-I) and IRF7 are upregulated in almost all cells in the TME, including epithelial cells, stromal cells and immune cells. Interestingly, one of the characteristics of cancer tissue after RT was found to be a heterogeneous immune response, as indicated by the presence of both hot and cold regions in a single tumour. In immunologically hot sites, the levels of several virus response signals, including DDX58 (RIG-I), were upregulated in conjunction with those of well-known signals such as IFN-γ and factors related to antigen presentation. Another characteristic was the prominent increase in the CD68 + HLD-DRA+ macrophage and DC populations compared with the CD4+, CD8+, and CD19+ lymphocyte populations, as determined in the ex vivo analyses. These results are consistent with previous literature reporting that irradiation elicits DC and macrophage activation[42]. Thus, we concluded that the LTR–RIG-I pathway would be a key regulator of macrophage and DC activation after irradiation. Recent scRNA-seq studies showed that DCs and macrophages collaborate with CD4 + CD8 + T cells in the antitumour immune reaction; thus, we speculate that DC and macrophage activation by IR would lead to T-cell activation and the antitumour immune reaction.

The newly discovered immune response regulated by the mTOR–LTR–RIG-I axis provides various therapeutic possibilities, and LTRs were reported to be activated by genetic factors and cancer treatment[13]. For example, a previous study showed that the LTR levels increased as a booster event upon treatment with DNMT, EZH2 or HDAC inhibitors. Currently, radiation

combined with chemotherapeutic drugs such as metabolic antagonists and platinum-based agents is cytotoxic, but combinations with chemotherapies that activate LTRs, such as Dnmt1 inhibitors, are expected to yield stronger immune responses. To date, the utility of numerous ICIs, molecularly targeted drugs, and oncolytic viruses, among other therapeutic modalities, has been verified in clinical trials and preclinical models exploring the potential of combination therapy involving immunotherapy and RT[43,44]. Our results suggest that MHY1487, a radiosensitizer and an mTOR activator[39], and Reolysin, an oncolytic virus derived from dsRNA[45] that is currently being studied in phase II and III trials for NSCLC and head and neck squamous cell carcinoma (NCT01708993 and NCT01166542), might be effective additions to combination therapy comprising immunotherapy and RT. Thus, elucidation of the molecular mechanism of the antitumour immune reaction is an important step in optimising RT combined with immunotherapy.

This study contains two major limitations.

One limitation is we focused on the LTR–RIG-I pathway, but we did not evaluate other candidate ligands and RNA sensor genes such as mtRNA and EIF2AK2. Furthermore, RIG-I and MDA5 were involved in the RNA sensor pathway; however, knockouts of these two genes showed different gene expression patterns, suggesting differences in their pathways. In particular, RIG-I recognises short RNA ligands (<300 bp) with 5′-triphosphate caps, and MDA5 recognises long dsRNA ligands (>1000 bp) with no end specificity. It is possible that irradiation could elevate different amounts of short/long dsRNA and lead to different loads on RIG-I or MDA5 signalling. To completely understand these mechanisms, we have to investigate future work.

Another limitation is the small sample size and use of only one sample per patient, except in the case of one patient, for the analysis. Endoscopy performed during or immediately after RT is not a standard protocol due to the risks of mucous membrane inflammation and bleeding; therefore, patient numbers were limited. Few biopsy samples collected during RT satisfied the sample quality requirements for scRNA-seq because of the tissue damage caused by irradiation; thus, a sufficient analysis could not be performed. Further studies are needed to investigate the immune features of patients after RT. The results of our clinical specimen analysis were not statistically conclusive; however, we consider these results to provide important information, as they are consistent with those of the in vitro experiments. Thus, although there are many challenges with on-treatment and post-treatment tissue analysis, this approach provides the most important information regarding the TME and radiation schedule. This is the first report of patient tissue sample analysis by scRNA-seq and spatial transcriptome approaches during and immediately after RT; more studies must be performed in the future to confirm our preliminary findings.

In summary, we found that the irradiation-induced cellular immune response is dominantly regulated by LTR–RIG-I–MAVS in NSCLC and ESCC cell lines. The LTR–RIG-I pathway was also upregulated in ESCC patient tissues, and targeting this pathway is a potentially important addition to clinical strategies for immunotherapy combined with RT.

## Methods

**Cell lines and reporter gene assay**. The oesophageal squamous cell cancer (ESCC) cell line KYSE-450 was obtained from the Japanese Collection of Research Bioresources Cell Bank, National Institutes of Biomedical Innovation and maintained at 37 °C with 5% $CO_2$ in RPMI 1640 medium (Sigma–Aldrich, Saint Louis, MO, USA) supplemented with 10% FBS (Biowest, Nuaille, France). A549-Dual cells [wild-type (WT), RIG-I knockout (KO), MDA5-KO and MAVS-KO] were obtained from Invitrogen (Carlsbad, CA, USA). Cell culture and luciferase assays were performed according to the instructions for the A549-Dual cells. Cytotoxicity was assessed and normalised using the alamarBlue assay (Thermo Fisher Scientific).

**Irradiation**. Irradiation was performed as previously described in ref. [46]. Briefly, using a Clinac® iX linear accelerator and a 6 MV photon beam, A549 cells and KYSE-450 cells were irradiated at a dose rate of 8 Gy/min. Nonirradiated (NIR) cells served as controls.

**Phosphoproteome analysis**. A 500-µg sample of protein extract from A549 cells was reduced, alkylated, and digested with trypsin/Lys-C mix as previously described in ref. [47]. The digested sample was desalted by using a MonoSpin C18 column (GL Sciences, Tokyo, Japan) and then dried with a centrifugal evaporator. Phosphopeptides were enriched using a Titansphere Phos-TiO kit (GL Sciences) according to the manufacturer's instructions and then analysed on an Orbitrap Exploris 480 with an UltiMate 3000 RSLCnano LC system (Thermo Fisher Scientific). MS data acquisition was performed in data-dependent acquisition (DDA)-MS mode and overlapping window data-independent acquisition (DIA)-MS mode[47]. In the DIA-MS mode used for quantification, the isolation width was set to 10 m/z with stepped normalised collision energies of 22, 26 and 30%. The isolation windows covered 400–1000 m/z, as optimised by Skyline 4.1[48]. In DDA-MS mode, to produce a spectral library, all samples were pooled for analysis using the gas-phase fractionation method. The MS ranges of 395–555, 545–705 and 695–1005 m/z were used in DDA mode. The spectral library of phosphopeptides was generated by searching the DDA-MS data against a human UniProt reference proteome (UniProt id UP000005640, reviewed, canonical) using Proteome Discoverer v2.3 (Thermo Fisher Scientific). Quantitative analysis of phosphopeptides was performed by Scaffold DIA v2.2.

**Immunocytochemistry assay (γ-H2AX)**. A549-Dual cells were grown on four-well chamber slides (Matsunami Glass, SCS-N04) for 24 h before irradiation at a dose of 8 Gy. At 3 h post-irradiation, the cells were fixed with 4% PFA in phosphate-buffered saline (PBS) for 15 min at room temperature, permeabilized with 0.1% Triton X-100 for 5 min, and blocked with 3% BSA in PBS for 30 min at 37 °C. Following overnight incubation at 4 °C with the primary antibodies anti-phospho-H2A.X (Ser139) (CST, #2577) and anti-dsRNA-K1 (CST, 28764), the cells were incubated with a DyLight 488-conjugated donkey anti-mouse IgG secondary antibody (Invitrogen, SA5-10166) for 30 min at 37 °C. The primary and secondary antibodies were diluted 1:100 and 1:250, respectively, in PBS containing 3% BSA. The cells were then mounted with VECTASHIELD mounting medium with DAPI (H1200, Vector Laboratories). Images were obtained with an all-in-one fluorescence microscope (BZ-X800, KEYENCE, Osaka, Japan) equipped with a Plan Apochromat 40x objective (NA0.95, BZ-PA40, KEYENCE, Osaka, Japan). Positive areas and signal intensities were automatically calculated using a hybrid cell count application (BZ-H4C, KEYENCE, Osaka, Japan) in BZ-X Analyzer software (BZ-H4A, KEYENCE, Osaka, Japan).

**Western blotting**. For protein extraction, cells were collected by scraping in cold PBS, washed with cold PBS, and then lysed in RIPA buffer (Wako, Japan) containing a protease inhibitor cocktail (Sigma, P8340) for 20 min on ice; the resulting lysate was cleared by centrifugation at 10,000×g and 4 °C for 10 min. The protein content was quantified by a BCA assay (Thermo Scientific), and equal amounts of protein from each sample were separated by SDS–PAGE (Wako, Japan) and transferred to a PVDF membrane (Bio-Rad). GAPDH was used as the loading control for normalisation. Bands were visualised using ImageQuant LAS4000 (GE technology). Images of western blots were obtained using ImageJ.

**RNA immunoprecipitation (RIP) assay**. A549-Dual cells were transfected with the pSF-3xFLAG-RIG-I vector using FuGENE-HD (Promega, E2311) following the manufacturer's instructions. After 24 h of transfection, the cells were irradiated at a dose of 8 Gy. The RIP assay was performed with a Magna RIP Kit (Millipore, 17-701) at 48 h post-irradiation. Briefly, protein A/G magnetic beads were washed and resuspended in RIP wash buffer. Five micrograms of anti-FLAG-M2 antibody (Sigma, F1804) was bound to 100 µL of beads for 30 min at room temperature; then, the conjugated beads were washed three times with RIP wash buffer. Treated cells from two 10-cm² plates were washed with 10 mL of cold PBS. The cells were scraped from the plate, transferred to a tube, and pelleted by centrifugation at 1500 rpm and 4 °C for 5 min. The cell pellets were lysed in 100 µL of RIP lysis buffer on ice for 5 min, followed by a single freeze–thaw cycle at −80 °C. The resulting lysate was centrifuged at 14,000 rpm and 4 °C for 10 min; 10% of the RIP lysate supernatant was removed as the input sample. For immunoprecipitation, the remaining supernatant was mixed with the bead–anti-FLAG-M2 antibody complex in RIP immunoprecipitation buffer and incubated overnight at 4 °C with rotation. Following magnetic separation, the beads were washed 6 times with 500 µL of cold RIP wash buffer. After protein digestion by proteinase K for 30 min at 55 °C, total RNA (input) and RIG-I-bound RNA were purified by the phenol–chloroform method. The quality of the extracted RNA was assessed on an Agilent TapeStation 2200 system, and RNA sequencing (RNA-seq) was performed.

**RNA purification and bioinformatics analysis**. For RIP RNA-seq, total RNA-seq was performed by Macrogen (Tokyo, Japan) with a SMARTer stranded kit (Clontech Laboratories). Sequencing data were subjected to quality control filtering, trimming and adaptor removal using FastQC and Trimmomatic[49]. All filtered sequences were aligned to the hg38 reference genome, and gene expression was represented as transcripts per million (TPM) calculated by RSEM[50]. Differentially expressed genes (DEGs) were identified by using the R package edgeR[51] with a false discovery rate (FDR) ≤0.05. DEGs were used for Gene Ontology (GO) and Kyoto Encyclopaedia of Genes and Genomes (KEGG) analyses with the R package clusterProfiler[52]. TE expression analysis was performed using the REdiscoverTE pipeline[19]. Heatmaps were drawn using the R package pheatmap.

**LTR transfection**. WT and RIG-KO A549-Dual cells were plated in a 96-well plate 48 h before transfection with 20 ng/well in vitro-synthesised 5′ppp-RNA or 5′cap-RNA encoding LTR21B (FASMAC, Kanagawa, Japan) using FuGENE-HD. Negative control cells were exposed to only the transfection reagent for 48 h after plating.

**Kinase inhibitor screening**. We performed kinase inhibitor screening with an ISRE reporter assay system and A549-Dual cells. The screening was performed in a 96-well plate using a semiautomatic INTEGRA VIAFLO 96 system (INTEGRA Biosciences, Tokyo, Japan). At 1 h before 8 Gy irradiation, the test inhibitors were added at a final concentration of 5, 0.5 or 0.05 μM. The reproducibility of the results was confirmed in at least two independent experiments. The results for the experimental nonirradiated (NIR) and irradiated (IR) groups (NIR or IR cells treated with an inhibitor) are shown as the fold change in comparison to the respective control cells (NIR or IR cells treated with only vehicle). Cytotoxicity was assessed using the alamarBlue assay, and the results were normalised accordingly. Concentrations of compounds that yielded a cell survival rate of 10% or less were excluded from further analysis.

**PBMC migration assay**. One millilitre of medium from IR and NIR A549 cells was transferred to each well (lower chamber) of a 24-well plate, and a ThinCert insert (upper chamber; pore size of 3.0 μm, 33.6 mm²; Greiner, #662631) was placed in each well. A total of 300,000 PBMCs (#PB009C-2, HemaCare, California) (Lot: 20062735) were diluted in 300 μL of RPMI medium supplemented with 10% FBS and then placed in the upper chamber; the plates were incubated at 37 °C for 4 h. Then, the insert was washed twice with PBS, stained with crystal violet, and observed under a microscope. The PBMCs that migrated through the insert and detached into the medium in the lower chamber were automatically counted on Countess™ Cell Counting Chamber Slides (Invitrogen). The experiment was reproduced with biological duplicates. RNA-seq was performed using WT and KO cells (RIG-I-KO, MAVS-KO) 16 h after irradiation or control (no irradiation) treatment. The immune score of each group (WT, RIG-I-KO and MAVS-KO) was evaluated with xCell[27].

**Single-cell RNA sequencing (scRNA-seq)**. Endoscopic biopsies were performed before and during RT to extract tissue samples from a minimum of two different points in accordance with the approved study protocol (IRB-2018-101). All relevant ethical regulations were followed. Tissue dissociation into single cells for scRNA-seq was performed with a Tumour Dissociation Kit (130-095-929; Miltenyi Biotec) following the default protocol in the user guide of the Chromium Single-Cell 5′ Reagent Kit (ver. 1,10x Genomics). Briefly, each biopsy sample was minced using Iris scissors and digested for 30 min at 37 °C with enzymes. The cell suspension was filtered through 70- and 30-μm strainers to remove cell aggregates and resuspended in PBS. scRNA-seq libraries were prepared using a Chromium Single-Cell 5′ Reagent Kit ver. 1 (10x Genomics) and sequenced using the HiSeq 3000 platform (Illumina). Cell Ranger (ver. 3.0.0) was used with default parameters to process the reference genome alignment and quantify cells and transcripts. The raw sequencing reads were mapped to the human genome assembly GRCh38. TE levels based on the scRNA-seq data were further quantified with the scTE pipeline[53] using the BAM files generated by Cell Ranger.

The single-cell expression data generated by scTE were further analysed with the R package SeuratV4[54]. Cells with a gene number >5000 or <200 or a mitochondrial gene ratio >10% were excluded from downstream analysis. Five scRNA-seq samples were collected, and the data were integrated using the sctransform pipeline in Seurat. The integrated object was then normalised and scaled. Next, principal component analysis (PCA) (RunPCA) was performed. The top 30 principal components (PCs) were selected and submitted to FindNeighbors, FindClusters, and UMAP (RunUMAP) to obtain clusters. Cell type was annotated manually by referring to published papers and by SingleR (ver. 1.6.1)[55]. The annotated cell clusters were verified using a known cell marker list. Immune cell clusters (macrophages, DCs, neutrophils, T cells, and B cells) and epithelial cells were selected for comparative analysis.

**10x Visium spatial transcriptomic analysis**. To analyse the TME after RT by using the FFPE Visium spatial gene expression assay (10x Genomics), we selected a 6.5 × 6.5 mm² optical region of a formalin-fixed paraffin-embedded (FFPE) tumour block with DV200 ≥ 50% that contained cancer cells, stromal cells, and immune cells. Visium spatial gene expression slides and reagent kits were used according to

the manufacturer's instructions (10x Genomics). Each capture area contained 5000 barcoded spots of 55 mm in diameter (100 mm centre-to-centre spacing between spots), providing an average resolution of 1 to 10 cells. FFPE tumour samples were prepared according to the recommended protocols (Tissue Preparation Guide, CG000408). Haematoxylin and eosin (H&E) staining and imaging were performed according to the protocol (Deparaffinization, H&E staining, Imaging and De-crosslinking, CG000408), with the exception that an alcoholic eosin solution was manually applied to yield an image with lighter haematoxylin staining. TruSeq Illumina libraries were generated and sequenced on a NovaSeq system (Illumina) at a minimum sequencing depth of 50,000 read pairs per spot by IntegraGen (Evry, France). Sequencing was performed with the recommended protocol (read 1: 28 cycles; i7 index read: 10 cycles; i5 index read: 10 cycles; and read 2: 50 cycles), yielding between 150 million and 224 million sequenced reads. The Space Ranger pipeline was used to process the Visium spatial gene expression data, including data generated from demultiplexing, hg38 human reference genome alignment, tissue detection, fiducial detection, and barcode/UMI counting, following the guidelines provided by the manufacturer.

The Visium spatial expression object was further analysed as follows to determine clusters and perform gene expression analysis. Only high-quality spots (≥180 genes/spot) were selected for subsequent analyses. The SCTransform function in Seurat was used for normalisation, and PCA (RunPCA) was used for dimensionality reduction. The top 30 PCs of each spot were used for graph-based clustering analysis, and the identified clusters were plotted by UMAP. Based on morphological characteristics and the expression of immunity-related marker genes, the spots were manually selected and annotated as immunologically active (hot) or suppressed (cold) sites. The FindMarkers function in Seurat was used to identify DEGs between the hot and cold sites using the criteria P ≤ 0.05 and fold change ≥2. The identified DEGs were used for GO enrichment analysis performed with the R package clusterProfiler.

**Statistics and reproducibility**. Statistical analysis of the dual-reporter assay (Figs. 1b, 3d, e and 5d), dsRNA staining (Fig. 2b), comparison of gene expression (Fig. 5c and Supplementary Fig. 4a), PBMC migration assay (Fig. 6c) data was performed with Excel (Microsoft). The statistical significance of differences between the two groups was assessed by an unpaired two-tailed Student's t-test. A p value <0.05 was considered to indicate statistical significance. Data were presented as the mean ± SEM.

The reproducibility of experiments was described in the legend of each figure. Briefly, dual-reporter assays (Figs. 1b, 3d, e, 5d and Supplementary Figs. 3e), dsRNA staining, comparison of gene expression (Figs. 3c, 5c and Supplementary Fig. 4a), PBMC migration assay were performed with biological replicates n = 3; inhibitors library dual-reporter assays (Fig. 3b and Supplementary Fig. 3a, b) were performed with biological replicates n = 2; RIP-seq (Fig. 3b), gene expression of time-course (Fig. 5b), immune score analysis was performed n = 1.

**Reporting summary**. Further information on research design is available in the Nature Portfolio Reporting Summary linked to this article.

## Data availability

The RNA-seq and scRNA-seq data relevant to the current study are available from DDBJ Sequenced Read Archive under the accession numbers PRJDB15356; the original of the western plot images presented in the main figures could be found in Supplementary Figs. 9,10; source data of the graphs and charts presented in the main figures could be found in Supplementary Data 1; mass spectrometry-based proteomics data could be found in 10.5281/zenodo.8024935; plasmid generated in this study is available upon request.

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

## Acknowledgements

This study was funded in part by Grants-in-Aid for Scientific Research KAKENHI [grant number 18K07740], the Japan Agency for Medical Research and Development (AMED) [grant numbers 18ck0106210h0003 and 19ck0106485h001] and the National Cancer Center Research and Development Fund [31-A-10]. The funders had no role in the study design and collection, data analysis and interpretation, writing of the manuscript or decision to submit the article for publication. We would like to express our appreciation for the help provided by the radiological technologists and medical physicists at National Cancer Center Hospital East.

## Author contributions

S.-I.K., M.O., J.D., M.N., H. Hojo, H. Hirata and A.M. conducted the experiments and radiotherapy. S.-I.K., Y.H., T.A., K. Tanaka and K. Tsuchihara wrote the manuscript. J.D., A.S., Y.S. and R.Y. performed the bioinformatics analysis. H.Hojo, J.D. and S.-I.K. performed the data analysis and prepared the figures for the manuscript. H.S. and T.Y. conducted an endoscopic biopsy. D.K., T.K., M.K. and Y.N. determined clinical diagnoses, patient management and administered treatment.

## Competing interests

The authors declare no competing interests.

## Consent to participate

Clinical specimens were collected from patients with squamous cell carcinoma of the oesophagus as part of an ongoing study. All analyses were performed with patient consent and IRB approval from the National Cancer Center East Hospital (IRB2018-101).
