## [Peer Review File · Communications Biology]

Reviewers' comments:

Reviewer #1 (Remarks to the Author):

The manuscript by Du et al. reported the molecular mechanism of radiotherapy-induced immune reactions. They showed that exposure to radiation results in the induction of LTRs through the mTOR pathway. Interestingly, they found that LTRs activate immune response via RIG-I rather than MDA5 activation. The authors further performed scRNA sequencing and spatial transcriptome to strengthen the involvement of LTRs and RIG-I during radiotherapy.

Overall, this paper presents thorough profiling showing the association among mTOR-LTR-RIG-I regulatory axis during radiotherapy. The manuscript is well written and the data presentation is clear. Despite the massive amount of data, few of the conclusions by the authors need further validation. Specifically,

- 1) The authors focused on the RIG-I pathway, but LTRs can also activate PKR. In addition, a recent study by Kim et al. (Cell Reports, 2022) showed PKR phosphorylation when chondrocytes were under acute irradiation. What is the role of PKR in this context?
- 2) On Line 136, the authors showed the correlation between RIG-I/MDA5 with several ISGs. However, RIG-I and MDA5 are also induced downstream of interferon response so focusing on this correlation and using it as a rationale for studying RIG-I is too strong.
- 3) Is there a specific group of LTRs that show strong interaction with RIG-I? In other words, what fraction of LTRs induced by radiotherapy showed strong binding to RIG-I? What are their sequence or structural characteristics?
- 4) As the authors pointed out, mTOR inhibition triggers autophagy which can affect the antiviral signaling by LTRs. What happens when autophagy is triggered through an mTOR-independent manner, like using metformin?
- 5) Line 208, EIF2AK3 is not an RNA sensor. Maybe, EIF2AK2?
- 6) It is interesting that MAVS knockout showed a similar transcriptional pattern as RIG-I knockout, but different from that of MDA5 knockout. Can authors elaborate more on this?
- 7) Line 330, "RIG-I showed a ligand preference for LTRs over other TEs". This statement is not supported. In this context, RIG-I showed strong interaction with LTRs because LTR expression was induced by the therapy.

Reviewer #2 (Remarks to the Author):

Radiotherapy (RT) is well known to induce antitumour immunological response in cancer. A number of studies have demonstrated the mechanism underlying this response, including interferon and MHC I upregulation. In this study, the authors examine the potential role of endogenous retrotransposable elements (TEs) in immunological effects induced by radiotherapy. The author first uses a lung cancer cell line RT model to demonstrate the upregulation of ISRE activity, which was found to coincide with increased activity in the RNA sensing pathway and increased dsRNA and LTR expression. RIP-seq using RIG-I further confirmed that LTRs are the most significant targets in irradiated cells. To further elucidate the mechanism, high-throughput screening of kinase inhibitors was performed to identify mTOR as a regulator of ISRE activity. Specifically it was found that mTOR inhibitor or KO had reduced ISRE activity which was accompanied by reduced TE expression. Further KO experiments of RIG-I and MDA5 confirmed that RT-induced immune response is dependent on these RNA sensing pathways. Finally, as post-RT lung biopsy is difficult to obtain, the authors used tissue from ESCC and performed scRNA-seq to confirm post-RT LTR upregulation and RIG-I activation. Spatial transcriptomics was also performed on one patient to identify sites of immune cell infiltration and damaged cancer cells.

Overall the study contains a substantial amount of novel data and uses a wide range of cutting-edge experimental techniques. The data analysis is sound and the conclusions are generally well supported. The novel finding that induction of LTR may underly RT response is important, although the study also leaves many unanswered questions, such as the role of mTOR in upregulating LTRs. Nonetheless, the findings are significant and will be an important reference for further research in the field.

A few specific comments:

(1) 8 Gy is quite a high dose for cell line treatment. It is surprising not to see enrichment of any DNA damage response pathways in Fig 1e. Although the authors confirm the relationship between dsRNA sensing and immune response in public TCGA data. It would be useful to confirm the upregulation of LTR in other published datasets of IR-treated cells, such as <https://www.ncbi.nlm.nih.gov/bioproject/PRJNA642705>

(2) ADAR1 is known to prevent innate immune sensing by editing dsRNA, which attenuates RIG-I activation. Can the authors show that ADAR1 and RNA editing levels are generally lowered in IR treated cells? It would be nice to see if ADAR1 overexpression can reduce ISRE activation, although this is probably out of scope of the current study.

(3) ReDiscoverTE only provides expression of intergenic TEs at the subfamily level. Other tools such as Telescope can provide an estimation of expression of individual TEs using EM algorithm. This may help provide clues regarding specific copies of TEs that are responsible for RIG-I binding and IRSE activation?

(4) The downregulation of TEs following mTOR-KO is interesting. Can the authors comment on whether the observation may be due to more rapid degradation/editing of dsRNA rather than increased regulation?

(5) The datasets are only available upon request to the authors. It will be important for these data to be deposited in public repositories such as GEO.

Reviewer #3 (Remarks to the Author):

This study found the important role of RIG-1 mediated virus sensing pathway in radiotherapy induced cellular immune reactions. Although it is an interesting topic, most of the data is not convincing enough as in most part of the manuscript, only the correlation of different genes are shown, the functional and mechanistic studies are very limited. For example, authors did not do any experiments to rule out the possibility that DNA sensing pathway is also important for radiotherapy -induced cellular immune reaction; There is no experiments proving RNA sensing pathway affecting tumor growth either in vitro or in vivo; And there is no study showing how this pathway is activated under irradiation. So I think more detailed functional study should be done before this study can be published.

I also have some other minor suggestions:

1. Page 5, MCF10A is not a breast cancer cell line. If normal cell lines end up having same responses as cancer cell lines, I am wondering if ncRNA-RIG-1-MAVS pathway would actually start to hurt normal cells. Would prefer to confirm if RIG-1 inhibitor combined with irradiation will have strong side effect or not in vivo.

2. Figure 1e, can authors address what is the x-axis?

3. Figure 2a, I am not sure if rH2AX staining is specific. Normally you should observe rH2X foci after DNA damage stimulation, but from the panel I don't see any.

4. Figure 3a, I think it would be more accurate to do endogenous RIP to assess the binding.

5. Supplementary fig 1a, please also provide the correlation plot and the p value.

6. Since CD8+ T and DC cells infiltration is correlated with RIG-1 expression, does RNA sensor pathway also induce adaptive immune response after radiation? Would suggest to do some experiments to distinguish that.

7. Figure 6a, except for migration assay, it is also important to show if PBMC would have strong killing effect towards RIG-1 KO or MAVS KO cells. It is also important to see if PBMC also has stronger killing effect on normal cells such as MCF10A with RIG-1 KO or RIG-1 inhibition. If it turns out that there is no selection between cancer cells and normal cells, I am worried the significance of this discovery.

8. What is the physiological function of upregulation of LTRs? And how does it affect immune response under irradiation?

Referee #1: immune response, therapy, viral RNAs

1) The authors focused on the RIG-I pathway, but LTRs can also activate PKR. In addition, a recent study by Kim et al. (Cell Reports, 2022) showed PKR phosphorylation when chondrocytes were under acute irradiation. What is the role of PKR in this context?

Thank you for your question.

In particular, PKR mainly responds to mitochondrial stress, while it regulates ISGs together with other dsRNA sensors, such as MDA5 and RIG-I. Due to the limitation of sensitivity, in our research, we could not assess PKR phosphorylation by LC-MS/MS, but a study by Hao et al. (*Oncotarget*, 2016) showed that radiation did not increase the expression of PKR or its phosphorylation in the A549 human lung cancer cell line.

In summary, we think PKR might also play an important role together with RIG-I in regulating the innate immune response under irradiation, but their interaction might be different between cell lines. We will keep an interest in this mechanism for future research.

2) On Line 136, the authors showed the correlation between RIG-I/MDA5 with several ISGs. However, RIG-I and MDA5 are also induced downstream of interferon response, so focusing on this correlation and using it as a rationale for studying RIG-I is too strong.

Thank you for your comment.

We edited Lines 137-139 to improve the readability and logical flow. Briefly, based on the results of LC-MS/MS, we found that pathways sensing RNA viruses and triggering the IFN response were activated after irradiation. Because there have been several studies showing that the irradiation-induced immune response is RNA sensor pathway dependent, we hypothesized that the immune response was associated with RIG-I/MDA5—MAVS in this context. The correlation of RIG-I/MDA5 with ISGs in LUAD tissue confirmed our hypothesis.

Additionally, the potential of using RIG-I as a novel cancer therapy is another rationale for studying this molecule. Recently, emerging interest has arisen regarding the use of targeting RIG-I signalling in oncology therapies (reviewed by Solstad, et al. *J Immunol*, 2022). In addition, several clinical studies have pointed out the advantages of activating the RIG-I/MDA5 pathway to enhance the effect of ICIs in cancer immunotherapy (reviewed by Chen, et al. *Sig Transduct Target Ther*, 2020).

Collectively, based on the LC-MS/MS results and previous studies, we decided to study the RIG-I pathway, and we believe that studying RIG-I can provide clues to develop new therapeutic opportunities for cancer radioimmunotherapy.

3) Is there a specific group of LTRs that show strong interaction with RIG-I? In other words, what fraction of LTRs induced by radiotherapy showed strong binding to RIG-I? What are their sequence or structural characteristics?

Thank you for your question.

In particular, RIG-I preferentially recognizes viral-like elements by an uncapped 5' end structure rather than a specific sequence. From our RIP-seq data, we could not conclude whether a specific group of LTRs showed a strong interaction with RIG-I.

To the best of our knowledge, the expression of some LTRs in tumours is associated with immune infiltration and increased antigenicity (Kong, Y. et al. *Nat Commun*, 2019); thus, we speculated that the regulation of the RIG-I pathway is more affected by the total expression level of LTRs than by interactions with specific groups.

4) As the authors pointed out, mTOR inhibition triggers autophagy which can affect the antiviral signaling by LTRs. What happens when autophagy is triggered through an mTOR-independent manner, like using metformin?

Thank you for your interesting question.

In particular, autophagy can suppress RIG-I-mediated interferon production. In our research, we treated A549 cells with three kinds of autophagy inhibitors and assessed ISRE activity. Compared to those treated with mTOR inhibitors, the decrease in ISRE activity caused by autophagy inhibitors was mild (Supplementary Fig. 3a, also shown below).

Collectively, we speculated that autophagy has a weak influence on the antiviral signalling described in our research.

5) Line 208, EIF2AK3 is not an RNA sensor. Maybe, EIF2AK2?

Thank you for pointing this out.

Because we do not have LC-MS/MS data for EIF2AK2, we deleted this part in the manuscript and edited Supplemental Figure 3b. Please check the highlighted Line 209 in the revised manuscript.

6) It is interesting that MAVS knockout showed a similar transcriptional pattern as RIG-I knockout, but different from that of MDA5 knockout. Can authors elaborate more on this?

Thank you for your comment.

We speculated that the different transcriptional patterns observed in MDA5-KO might stem from two mechanisms:

1. In particular, RIG-I recognizes short RNA ligands (<300 bp) with 5'-triphosphate caps, and MDA5 recognizes long dsRNA ligands (>1000 bp) with no end specificity. It is possible that irradiation could elevate different amounts of short/long dsRNA and lead to different loads on RIG-I or MDA5 signalling.

2. Studies from Feng Xu et al. (*The EMBO journal*, 2020) and Tigano Marco et al. (*Nature*, 2021) showed that the IR-induced RNA-sensing pathway was mainly associated with RIG-I. To the best of our knowledge, there are few reports regarding the MDA5 dependency of IR-induced RNA sensing. Based on our data, the induction of RIG-I expression under irradiation was stronger than that of MDA5, which indicated that RIG-I might be the dominant response molecule in this context; thus, the KO of MDA5 had a weak influence downstream.

Collectively, in this study, we found that LTR-RIG-I/MAVS signalling is the major pathway. We would be interested in studying the mechanism of MDA5 in the future.

7) Line 330, “RIG-I showed a ligand preference for LTRs over other TEs”. This statement is not supported. In this context, RIG-I showed strong interaction with LTRs because LTR expression was induced by the therapy.

Thank you for your comment.

We edited the manuscript at Line 331 as follows: “RIG-I showed a strong interaction with the RT-induced LTRs.”

Referee #2: TEs, immune evasion, cancer

(1) 8 Gy is quite a high dose for cell line treatment. It is surprising not to see enrichment of any DNA damage response pathways in Fig 1e. Although the authors confirm the relationship between dsRNA sensing and immune response in public TCGA data. It would be useful to confirm the upregulation of LTR in other published datasets of IR-treated cells, such as <https://www.ncbi.nlm.nih.gov/bioproject/PRJNA642705>

Thank you for your comment and suggestion.

1. We found that the protein phosphorylation level of CHEK2 was increased after irradiation (Supplementary Figure 3b, also shown below), indicating that the DNA damage response was induced, although we could not find enrichment of the DNA damage response.

Additionally, as far as we know, 8 Gy~ has been used in cell line experiments in several studies related to the radio-induced immune response. Examples include the A549 human lung cancer cell line (20 Gy, Tigano et al., *Nature*, 2021) and MCF10A breast fibroblast cell line (20 Gy, Harding et al., *Nature*, 2017).

2. As you suggested, we applied REDiscoverTE to the datasets you mentioned. We confirmed the upregulation of a subset of LTRs and RIG-I/MDA5 in A549 cells treated with 6 Gy of X-ray IR (Figure below, left). We also confirmed the upregulation of RIG-I and several ISRE genes in this dataset (Figure below, right).

(2) ADAR1 is known to prevent innate immune sensing by editing dsRNA, which attenuates RIG-I activation. Can the authors show that ADAR1 and RNA editing levels are generally lowered in IR treated cells? It would be nice to see if ADAR1 overexpression can reduce ISRE activation, although this is probably out of scope of the current study.

Thank you for your inspiring question.

1. We calculated the RNA editing (A to I) level using the original script. The workflow is shown below.

Briefly, bam files of IR and non-IR RNA-seq were used as input of samtools (<http://www.htslib.org>) to generate vcf files. The shared mutated positions in IR and NIR (IR_NIR_overlapped) were used in the following calculation. Valid positions of A-to-I editing were downloaded from REDiportal (<http://srv00.recas.ba.infn.it>). Total A-to-I positions (-Total) were further subset as non-Alu positions (-rmALU) and LTR only position (-LTR) in the following calculation.

RNA editing levels were estimated by the proportion of positions with A-to-I editing (A to G, AtoG_position column) within the total/rmALU/LTR mutated position (rows: all, -Alu, LTR) in IR and NIR, respectively. The ratio of AtoI_position/IR_NIR_overlapped was used to represent the RNA editing level.

We found that the RNA editing level was similar between non-IR- and IR-treated cells, as shown in the table below. Thus, we considered that RNA editing had limited effects on ISRE activation in this context.

		G			H	
		REDiportal record	BAM hit	IR_NIR_overlapped	AtoG_position	H/G
IR	all	15,681,871	5,645,835	17216	633	3.68%
IR	- Alu	1,465,173	362,019	9424	412	4.37%
IR	LTR	235,064	33,399	67	1	1.49%
NIR	all	15,681,871	5,797,887	17216	645	3.75%
NIR	- Alu	1,465,173	369,331	9424	367	3.89%

2. Our results showed that the gene expression of ADAR was upregulated in the A549 human lung cancer cell line after IR treatment (Figure below). This result was consistent with a study from Komatsu et al. (*Science report*, 2022), which showed that ADAR1 expression was upregulated in the HT29 human colon adenocarcinoma cell line under IR treatment.

In summary, we think RNA editing or ADAR has limited influence on ISRE activation in this context.

(3) ReDiscoverTE only provides expression of intergenic TEs at the subfamily level. Other tools such as Telescope can provide an estimation of expression of individual TEs using EM algorithm. This may help provide clues regarding specific copies of TEs that are responsible for RIG-I binding and IRSE activation?

Thank you for your helpful suggestion.

As you suggested, we applied Telescope to our RIP-seq data. Briefly, Bowtie2 was used to generate bam files from RIG-I RIP and input RNA-seq data. Bam files of RIG-I RIP and input RNA-seq were used as the input of the telescope. Genome references, including the loci of all TEs downloaded from Repeatmasker (<https://www.repeatmasker.org/species/hg.html>), were used as references for Telescope. All parameters of Telescope and Bowtie2 were applied as default.

Focusing on LTRs, we compared the fold change (RIP/Input) of each individual TE proportion, and we found that 17 LTRs had relatively high enrichment ($\log_2FC > 1$) in RIP samples, as shown in the table below. Nevertheless, we found that most of the read counts of individual LTRs were small (less than 50 read counts, 0.01% of total read counts). Due to the limited read counts of our RIP-seq data, in this context, we think it is difficult to conclude that specific copies of TEs are responsible for RIG-I binding and IRSE activation. We will keep this question in mind for future research.

transcript_id	repClass	repfamily	repname	log2FC_prop	final_prop.RIP	final_prop.input	seqnames	start	end	width	strand
MLT1C_dup19325	LTR	ERV1-MaLR	MLT1C	40.77	2.16E-05	1.15E-17	chr21	42883435	42883845	411	+
MER4-int_dup2813	LTR	ERV1	MER4-int	31.44	5.35E-05	1.84E-14	chr22	35663672	35664338	667	+
ERV1-B4-int_dup1809	LTR	ERV1	ERV1-B4-int	18.96	7.11E-11	1.39E-16	chr8	23035442	23036562	1121	+
MLT1K_dup15687	LTR	ERV1-MaLR	MLT1K	17.35	2.91E-06	1.74E-11	chr19	17268043	17268226	184	-
MLT1J_dup14258	LTR	ERV1-MaLR	MLT1J	8.74	8.51E-05	0.00000199	chr20	43585052	43585530	479	-
LTR8_dup181	LTR	ERV1	LTR8	7.39	7.64E-05	0.00000456	chr1	2.02E+08	2.02E+08	726	-
LOR1-int_dup1058	LTR	ERV1	LOR1-int	6.41	2.59E-05	0.00000304	chr14	32268737	32269744	1008	+
MER21C_dup4754	LTR	ERV1	MER21C	6.24	1.97E-08	2.61E-10	chr17	56887680	56888380	701	-
MER52D_dup347	LTR	ERV1	MER52D	5.41	0.0025	0.0000589	chr9	81689313	81689527	215	-
MLT1B_dup14252	LTR	ERV1-MaLR	MLT1B	4.83	0.000397	0.000014	chr14	31446490	31446633	144	-
MST-int_dup202	LTR	ERV1-MaLR	MST-int	4.31	2.57E-05	0.0000013	chr6_GL000251v2_alt	1973442	1973808	367	+
LTR40a_dup1284	LTR	ERV1	LTR40a	3.53	5.98E-05	0.00000518	chrX	90311199	90311678	480	-
HERVE-int_dup257	LTR	ERV1	HERVE-int	3.14	5.76E-05	0.00000653	chr19	28607468	28613524	6057	+
MLT1H_dup6004	LTR	ERV1-MaLR	MLT1H	3.00	1.36E-11	1.7E-12	chr10	6737298	6737688	391	+
LTR8_dup2123	LTR	ERV1	LTR8	2.07	2.70E-13	6.44E-14	chr10	73742860	73743582	723	-
LTR19A_dup409	LTR	ERV1	LTR19A	1.57	1.14E-05	0.00000384	chr19	43647834	43648180	347	-
LOR1-int_dup279	LTR	ERV1	LOR1-int	1.21	2.24E-05	0.00000967	chr3	44750216	44751234	1019	+

final_prop. = read count of each TE/total read count, $\log_2FC_prop = \log_2(\text{final_prop.RIP}/\text{final_prop.input})$

(4) The downregulation of TEs following mTOR-KO is interesting. Can the authors comment on whether the observation may be due to more rapid degradation/editing of dsRNA rather than increased regulation?

Thank you for your inspiring question.

As explained in comment (2), we think the RNA editing level has limited influence on the RNA-sensing signalling we described in this study.

In addition to RNA editing, in our study, the results of LC-MS/MS (Supplementary Figure. 3b, also shown below, upper graphic) showed that the phosphorylation levels of proteins related to LTR expression and LTR stabilization were increased after IR. In addition, inhibition of mTOR partially abrogated this increase. Thus, we hypothesized that mTOR indirectly regulates the expression and stabilization of TEs.

Collectively, we speculated a model in which both the expression and stabilization regulation originating from mTOR affect the expression of TE (graphic shown below, lower graphic).

(5) The datasets are only available upon request to the authors. It will be important for these data to be deposited in public repositories such as GEO.

Thank you for your comment.

We are processing data deposition and the datasets would be available on DDBJ. We would provide the url link as soon as we finish.

Referee #3: radiation and immunology

This study found the important role of RIG-1 mediated virus sensing pathway in radiotherapy induced cellular immune reactions. Although it is an interesting topic, most of the data is not convincing enough as in most part of the manuscript, only the correlation of different genes are shown, **the functional and mechanistic studies are very limited**. For example, authors did not do any experiments to rule out the possibility that DNA sensing pathway is also important for radiotherapy -induced cellular immune reaction; There is no experiments proving RNA sensing pathway affecting tumor growth either in vitro or in vivo; And there is no study showing how this pathway is activated under irradiation. So I think more detailed functional study should be done before this study can be published.

Thank you for your valuable comments.

As you note, this study does not fully explain the mechanism of the radiation-induced RNA sensor pathway. Our results reveal upstream ligands of radiation-induced immune responses and activation of these pathways in patient tissue. However, we consider that these results make a meaningful report.

1) Key immune-related genes such as MHC class I and PD-L1 are dominantly regulated by the RNA sensor pathway in the non-small cell lung cancer cell line. Previous reports have shown that the mechanism of the radiation-induced immune response differs for each cancer type. Non-small cell lung cancer is the only cancer adapted to ICI-combined radiation therapy, and this is an important finding to improve cancer therapy.

2) Although there have been many reports on RNA sensor pathways in recent years, there are few reports on ligands. We analysed direct RNA binding to RIG-I by RIP assay and found that TE is a radiation-induced ligand via the mTOR pathway. It has been reported that TE is involved in cancer immune responses, and we consider this to be a reasonable result.

3) Finally, we confirmed that TE and RIG-I are upregulated in patient tissues. There are no reports demonstrating radiation-induced upregulation of TE in patient tissues, and this is a new finding.

Thus, although the results from our study do not completely elucidate the mechanism, they make a valuable report because they contain important findings.

The abstract has been changed to clarify that the mechanism has not been fully elucidated, as pointed out by the reviewer (Page 3 Line 42).

This study found the important role of RIG-1 mediated virus sensing pathway in radiotherapy induced cellular immune reactions. Although it is an interesting topic, most of the data is not convincing enough as in most part of the manuscript, only the correlation of different genes are shown, the functional and mechanistic studies are very limited. For example, authors did not do any experiments to rule out the possibility that DNA sensing pathway is also important for radiotherapy -induced cellular immune reaction; There is no experiments proving RNA sensing pathway affecting tumor growth either in vitro or in vivo; And there is no study showing how this pathway is activated under irradiation. So I think more detailed functional study should be done before this study can be published.

The DNA sensor pathway and the RNA sensor pathway act simultaneously in radiation-induced immune responses, and their ratio differs from cell to cell. A paper reporting a RIG-I-dependent radiation-induced immune response indicated that the RNA pathway is dominant, but the DNA-dependent immune response partially functions ((*EMBO J.* 39, e104036 (2020), *Nature* 591, 477-481 (2021))). We also confirmed that the DNA sensor functions partially in experiments using DNA pathway inhibitors.

We conducted inhibitor experiments and investigated the dependency of DNA sensor pathways (Figure on the left). This result shows that the DNA sensor pathway has limited influence.

We agree that the interaction between the DNA pathway and RNA pathway is an important yet under-researched topic. However, because of the high level of common downstream targets, it was difficult for us to rule out the influence of the DNA sensing pathway on radio-induced cellular immune reactions.

This study found the important role of RIG-1 mediated virus sensing pathway in radiotherapy induced cellular immune reactions. Although it is an interesting topic, most of the data is not convincing enough as in most part of the manuscript, only the correlation of different genes are shown, the functional and mechanistic studies are very limited. For example, authors did not do any experiments to rule out the possibility that DNA sensing pathway is also important for radiotherapy -induced cellular immune reaction; **There is no experiments proving RNA sensing pathway affecting tumor growth either in vitro or in vivo;** And there is no study showing how this pathway is activated under irradiation. So I think more detailed functional study should be done before this study can be published.

We used amalar blue-normalized ISRE activity to indicate the immune response per cell line (Page 21 Line 439). Therefore, normalized values are presented without showing the growth rate.

The results showed that there was no difference in the growth rate among RIG-I, MDA-5, and MAVS knockout cell lines (figure below). All A549-KO cell lines are commercially available and have normal growth rates (A549 lung carcinoma | Human NF- \$\kappa\$ B & IRF Reporter Cells (invivogen.com)).

1. Page 5, MCF10A is not a breast cancer cell line. If normal cell lines end up having same responses as cancer cell lines, I am wondering if ncRNA-RIG-1-MAVS pathway would actually start to hurt normal cells. Would prefer to confirm if RIG-1 inhibitor combined with irradiation will have strong side effect or not in vivo.

Thank you for pointing out this issue. We edited the manuscript related to MCF10A (Page 5, Line 78). However, I apologize, but I could not confirm the report that MCF10A is a normal cell line.

“Fibrocystic Disease” at ATCC (<https://www.atcc.org/products/crl-10317>)

“Breast Cancer” at COSMIC (<https://cancer.sanger.ac.uk/cosmic/sample/overview?id=2318371>)

As pointed out by the reviewer, radio-induced immune response in mucous membranes, lungs, etc., commonly occurs as adverse events in radiotherapy, especially for lung cancer and oesophageal cancer. The damage to normal tissue by the immune response is thought to be similar to that to cancer tissue (*Nat Rev Immunol.* 2022 Feb;22(2):124-138). Therefore, we consider both cancer tissue and normal tissue, and the analysis of the RNA sensor pathway is important.

On the other hand, due to the expression of neoantigens, cancer tissue attracts more attacks from infiltrated immune cells than normal tissue, and this selectivity can be a therapeutic target. Based on your comment, we tried to detect differences in immune responses between normal and cancer tissues. In brief, we annotated cancer tissue and normal mucous membranes in the VISIUM data and compared gene expression. The brief workflow is shown in the figure below. We would appreciate it if you could understand that these results were obtained at a preliminary level and that the data are shown in a confidential manner.

As shown in the figure below, RIG-I (DDX58) and PD-L1 (CD274) show a higher level of expression in cancer cells, and RNA sensor-dependent immune responses were shown to be activated more strongly in cancer tissue.

2. Figure 1e, can authors address what is the x-axis?

Thank you for your comment.

We added the title of the x-axis in Fig. 1.e (also shown below) and added a description of the x-axis in the legend (Page 38, Line 786), which represents the count of differentially expressed genes (DEGs) belonging to each GO term.

e GO enrichment analysis of phosphor-peptides

3. Figure 2a, I am not sure if rH2AX staining is specific. Normally you should observe rH2X foci after DNA damage stimulation, but from the panel I don't see any.

Thank you for pointing this out.

To confirm the colocalization of dsDNA and DNA damage, we conducted rH2AX staining. However, there was no colocalization, and γ H2AX had no staining. As you comment, we decided to delete the rH2AX staining in the manuscript.

4. Figure 3a, I think it would be more accurate to do endogenous RIP to assess the binding.

Thank you for your suggestion.

We tried to recover endogenous RIG-I but unfortunately failed, which may be due to the low specificity of the RIG antibody we used. Currently, we cannot find any commercially available RIG antibody suitable for RIP assays. We will attempt endogenous RIP in the future.

5. Supplementary fig 1a, please also provide the correlation plot and the p value.

Thank you for your comment.

We redid Supplement Fig. 1a with the latest version of the TCGA dataset and added Supplement Fig. 1b as its correlation plot with the p value.

a

Correlation of expression in NSCLC tissues (Spearman's Correlation)

	RIG-I	MDA5	cGAS	IFNG
OAS2	0.699	0.744	0.274	0.257
MX1	0.651	0.662	0.162	0.06
OASL	0.499	0.578	0.345	0.359
IFNB1	0.304	0.294	0.305	0.153
IRF7	0.322	0.342	0.092	0.084
IRF9	0.539	0.471	0.156	0.259

6. Since CD8+ T and DC cells infiltration is correlated with RIG-1 expression, does RNA sensor pathway also induce adaptive immune response after radiation? Would suggest to do some experiments to distinguish that.

Thank you for your inspiring question.

As the reviewer pointed out, the T-cell adaptive immune response is very important, but it is difficult to induce priming and effector phases in vitro, and the HLA types of PBMCs and cell lines are different. We think that it is difficult to clearly evaluate adaptive immune reactions using cell lines in this context.

Therefore, in our other ongoing project, we decided to conduct tissue analysis in ESCC patients who received radiotherapy by performing integrated spatial analysis with HE staining, multiplex immunostaining (CODEX) and spatial transcriptome (VISIUM). The brief workflow is shown in the graphic below. We would appreciate it if you could understand that these results were obtained at a preliminary level and that the data are shown in a confidential manner.

As shown in the figure below, the CD4-CD8 interaction was observed only in irradiated cancer tissue, indicating the induction of the adaptive immune response. We believe that such integrated analysis would allow us to distinguish the role of the adaptive immune response after radiation in the future.

7. Figure 6a, except for migration assay, it is also important to show if PBMC would have strong killing effect towards RIG-1 KO or MAVS KO cells. It is also important to see if PBMC also has stronger killing effect on normal cells such as MCF10A with RIG-1 KO or RIG-1 inhibition. If it turns out that these is no selection between cancer cells and normal cells, I am worried the significance of this discovery.

Thank you for your comment.

As you mentioned, the radiotherapy-induced inflammatory response causes damage in normal tissue (figures below).

Because tissue damage caused by the immune response is the comprehensive result of innate and adaptive immune responses and in vitro experiments using normal lung/oesophageal epithelial cells are difficult, we conducted tissue analysis to verify the mechanism we found in cancer cell line experiments.

As the results shown to answer comment #1, we demonstrate the selectivity of RIG-I expression and CD4, CD8 T-cell infiltration, thus we considered that this discovery has clinical significance.

8. What is the physiological function of upregulation of LTRs? And how does it affect immune response under irradiation?

Thank you for your inspiring question.

Typically, radio-induced LTR induction is involved in biodiversity. (Miousse IR, et al. *Mutat Res Rev Mutat Res*. 2015). The dose used in the treatment of cancer is mostly more than 8 Gy. Since it is a lethal dose for the whole body, whether the increase in LTR at such a high dose has physiological significance is still unclear.

However, half of cancer patients receive > 8 Gy radiation treatment, which can be fatal in the context of whole body exposure; thus, we think it is important to investigate how cancer tissue and cancer cells respond to this radiation dose.

Nonradiation-induced LTR upregulation in cancer tissue has been found to be involved in the activation of cancer immune responses (*Nature Communications*, 2019). Several hypotheses have been speculated, such as the release of type I IFN and an increase in neoantigens, but these hypotheses have not yet been proven (figure below). We think it is a challenging topic to verify these hypotheses and to determine whether similar events will occur with radio-induced LTR activation. We will keep an interest in this topic for future research.

Nature Communications Volume 10,
Article number: 5228 (2019)

Reviewers' comments:

Reviewer #1 (Remarks to the Author):

Kageyama and colleagues submitted a revised version and provided a clear point-by-point response. However, this reviewer still has a number of outstanding concerns that are not addressed properly. Specifically,

1) Even though EIF2AK2 is not detected in LC-MS/MS, it doesn't mean that EIF2AK2 is not involved. The manuscript is written such that all the downstream effect is mediated by RIG-I. The limitation and the potential contribution by other dsRNA sensors need to be discussed.

2) It is unclear why the authors chose LTR21B as a read-out, especially because there are other LTRs with stronger induction. Please justify.

3) The conclusion for Figure 2e is not correct. Fig. 2e shows that the activity of LTR21B is RIG-I dependent. This does not prove that LTR is the main ligand for RIG-I.

4) Authors should have a discussion about the difference between MDA5 KO Vs. RIG-I KO. Just presenting possible explanations in the response letter is not sufficient. Also, what is the length distribution of LTRs that interact with RIG-I from RIP-Seq data? It is true that RIG-I recognizes shorter dsRNA than MDA5, but the authors never showed that LTRs that interact with RIG-I are short, and thus, wouldn't be able to interact with MDA5.

Reviewer #2 (Remarks to the Author):

Thank you for addressing all my concerns. I support the publication of the manuscript.

Reviewer #3 (Remarks to the Author):

Most questions have been well addressed. It is acceptable that some questions cannot be addressed due to technical obstacles. I just have one more question. For question 1, when authors tested the immune response of normal and cancer tissues, did cancer tissues suffer more damage than normal tissues (more cell death)?

Reviewer #1 (Remarks to the Author):

Kageyama and colleagues submitted a revised version and provided a clear point-by-point response. However, this reviewer still has a number of outstanding concerns that are not addressed properly. Specifically,

1) Even though EIF2AK2 is not detected in LC-MS/MS, it doesn't mean that EIF2AK2 is not involved. The manuscript is written such that all the downstream effect is mediated by RIG-I. The limitation and the potential contribution by other dsRNA sensors need to be discussed.

Thank you for your comments. We edited Line 411-420 on page 19 as followed:

This study contains two major limitations.

One limitation is we focused on LTR-RIG-I pathway, but we did not evaluate other candidate ligands and RNA sensor genes such as mtRNA and EIF2AK2. Furthermore, RIG-I and MDA5 were involved in the RNA sensor pathway, however, knockouts of these two genes showed different gene expression patterns, suggesting differences in their pathways. In particular, RIG-I recognizes short RNA ligands (<300 bp) with 5'-triphosphate caps, and MDA5 recognizes long dsRNA ligands (>1000 bp) with no end specificity. It is possible that irradiation could elevate different amounts of short/long dsRNA and lead to different loads on RIG-I or MDA5 signalling. To completely understand these mechanism, we have to investigate as the future work.

2) It is unclear why the authors chose LTR21B as a read-out, especially because there are other LTRs with stronger induction. Please justify.

Thank you for your comments. We edited Line 168-170 on page 9 as followed:

As LTR21B has been reported to correlate with local tumour immune responses^{13,21}, although it is not clear that LTR21B is main ligand in radiation induced immune reaction, we chose to focus subsequent experiments on this specific LTR as an indicator.

3) The conclusion for Figure 2e is not correct. Fig. 2e shows that the activity of LTR21B is RIG-I dependent. This does not prove that LTR is the main ligand for RIG-I.

Thank you for your comments. We edited this conclusion on Line 172, page 9:

Transfected LTR21B significantly induced ISRE activity from 24 h to 48 h (Fig. 3d), similar to the effect of irradiation. ISRE activity was not induced by LTR21B in RIG-I-KO cells (Fig. 3e), suggesting that activity of LTR is RIG-I dependent in A549 cells.

4) Authors should have a discussion about the difference between MDA5 KO Vs. RIG-I KO. Just presenting possible explanations in the response letter is not sufficient. Also, what is the length distribution of LTRs that interact with RIG-I from RIP-Seq data? It is true that RIG-I recognizes shorter dsRNA than MDA5, but the authors never showed that LTRs that interact with RIG-I are short, and thus, wouldn't be able to interact with MDA5.

1. As pointed out by the reviewers, the difference between MDA5 and RIG-I ligands is an interesting issue. Many studies have been reported on MDA5 and RIG-I ligands, which remains controversial. However, all reports agree that while RIG-I and MDA5 have common ligands, some of them are distinct. (Nat Rev Immunol. 2020 Sep;20(9):537-551., Front Immunol. 2019 Jul 17;10:1586.) Our results suggest that the RNA sensor pathway is dependent on both RIG-I and MDA5, while they are partially different, which is consistent with earlier findings.

2. Thank you for your suggestion for analysis of RNA recovered by RIP.

In our study, we attempted a detailed analysis of the RNA recovered by RIP-seq, but due to its extremely limited recovery amount, it proved challenging to do so (figure below, right panel). In addition, because the cDNA was generated using random primers, it was difficult to estimate the size of the ligand from the sequence data.

Reviewer #2 (Remarks to the Author):

Thank you for addressing all my concerns. I support the publication of the manuscript.

Thank you for your support. We uploaded our data in DDBJ and add the link in the Data availability section.

Reviewer #3 (Remarks to the Author):

Most questions have been well addressed. It is acceptable that some questions cannot be addressed due to technical obstacles. I just have one more question. For question 1, when authors tested the immune response of normal and cancer tissues, did cancer tissues suffer more damage than normal tissues (more cell death)?

Yes, we believe that cancer tissue is more susceptible to immune attack than normal tissue. Although the cytokine-derived innate immune reaction is equivalent, the presence of neoantigen is thought to result in a stronger antigen-antibody reaction. However, we believe that these are still hypotheses and need to be proven in the future.